# The effect of entrepreneurship education on the entrepreneurial intention of different college students: Gender, household registration, school type, and poverty status

Wanli Deng[1,2]*, Juan Wang[1]

**1** School of Economics, Guangxi University, Nanning, Guangxi Zhuang Autonomous Region, PR China,
**2** School of Economics and Management, Guangxi University of Science and Technology, Liuzhou, Guangxi Zhuang Autonomous Region, PR China

* 100000870@gxust.edu.cn

**Data Availability Statement:** All relevant data are within the paper and its Supporting Information files.

## Abstract

Entrepreneurship education (EE) is a crucial link to promoting college students' entrepreneurship, which reduces unemployment, economic development, and poverty. Based on a sample of Chinese college students, this study uses a logistic model to investigate the effect of EE on entrepreneurial intention (EI). It focuses on evaluating the impact of EE on the EI of different groups of college students from the perspectives of gender, household registration, school type, and poverty status. Benchmark regression results show that EE has a significant positive impact on the EI of students. The heterogeneity analysis has the following findings. First, EE has a more significant impact on women's EI, which can reduce the entrepreneurial gap between women and men. Second, EE is more effective in improving the EI of urban students, which will further widen the gap between urban and rural students in entrepreneurship. Third, EE has increased the EI of students from private universities, which represent application-oriented universities. This shows that public universities, which represent research-oriented universities, need to increase the training of talent in practical applications to narrow the gap with private universities in entrepreneurship. Fourth, after receiving EE, the EI of nonpoor students improved more than that of poor students. Equal EE increases the gap between poor and nonpoor students, which can easily lead to an intergenerational cycle of poverty in entrepreneurship. This study provides empirical evidence from college students' entrepreneurship in relatively underdeveloped western China, which supports the development of EE and entrepreneurial activities.

## Introduction

Higher education and entrepreneurial activity drive economic growth, employment, entrepreneurship, and poverty reduction [1–4]. Entrepreneurship education (EE) and training are policy tools to promote economic development and poverty alleviation through education [5–

**Funding:** This research was supported by the following funding projects: 2017 National Social Science Foundation of China (Grant No. 17BGJ024); 2022 Philosophy and Social Science Planning Research Project of Guangxi Zhuang Autonomous Region (Grant No. 22AJL001); 2020 Guangxi Higher Education Undergraduate Teaching Reform Project (Grant No. 2020JGZ174); 2022 Guangxi Vocational Education Teaching Reform Research Project of Guangxi Department of Education (Grant No. GXGZJG2022B074); 2022 Undergraduate Education and Teaching Reform Project of Guangxi University of Science and Technology (Grant No. 2022XJJG45). The funders had no role in study design, data collection, and analysis, publication decisions, or manuscript preparation.

**Competing interests:** The authors have declared that no competing interests exist.

17]. EE is growing globally [2, 18–32]. In the global innovation and emerging technology market environment, college students face fierce competition and difficulties in entrepreneurship [2]. Formal education is the decisive factor in students' entrepreneurial choice and success, but the mismatch with entrepreneurial demand forms an obstacle to entrepreneurship [27, 28, 33–37]. EE, which is closely related to entrepreneurship, can effectively solve the problem of mismatch between formal university education and entrepreneurship and entrepreneurial barriers and improve entrepreneurial intention (EI) and the success rate [2, 28, 38–40].

There have been many studies on the impact of EE on EI [20, 41]. Previous studies were mainly conducted from the perspectives of the overall impact of EE on EI [1, 19, 22, 25, 38, 42–49], gender differences [22, 39, 50–53], the economic situation [6, 13–17], human capital theory [15, 20, 54–57], self-efficacy theory [2, 13, 17, 49, 58–65] and planned behavior theory [12, 25, 34, 64, 66–72]. Although there is much literature on EE's overall impact on EI and the differences in gender and economic conditions, there are still some research gaps. First, there are few relevant studies on the urban–rural gap, school type difference, and even rarer ones based in China [73]. Although there have been studies on the differences in EIs between college students from urban and rural areas, few have assessed the impact of EE on urban–rural differences in EIs [35, 44, 74]. Moreover, the empirical conclusions of relevant studies on school types are inconsistent. Some scholars believe that EE has a better effect on the EI of students from public universities than private universities [39], while others hold the opposite view [53, 75]. Second, there is a lack of empirical evidence that has been rigorously evaluated and tested [22, 41, 73, 76]. With EE's rapid development, it is generally believed that education and training have a significant impact on entrepreneurship, but there is still a lack of rigorous evaluation and testing [3, 22, 33, 56]. The empirical results have inconsistencies and ambiguities [20, 42]. Data quality and empirical methodology may be responsible for the empirical findings of positive, small, and inverse relationships [20, 22, 33].

Based on the existing research gaps, this study's primary purpose is to evaluate and analyze the impact of EE on the EI of college students in general and in different groups. The core research question of this paper is as follows: What is the specific quantitative value of the impact of EE on college students' EI? Are there significant differences among different groups of college students? To answer this question, the research content of this study is based on the questionnaire data of Chinese college students, and the logistic model and Stata 17.0 econometric analysis software are used for regression analysis and empirical tests. While evaluating the overall effect of EE on EI, this study examines the differences between male and female college students, urban and rural settings, public and private universities and poor and nonpoor college students. In addition, the empirical results of this paper are compared with previous literature, and human capital theory, self-efficacy theory, and planned behavior theory are used for analysis.

This study fills in the gaps in the literature and makes the following contributions. First, this paper evaluates the impact of EE on the EI of different college students. Previous studies have assessed the overall impact and differences in gender and economic status, with few other differences examined. In addition to the heterogeneity analysis of gender and economic status, this study also examined the differences between urban and rural household registration and school types to fill in the gaps in previous studies. Second, it provides deterministic empirical evidence that has been rigorously tested for the impact of EE on EI. To reduce the adverse impact of endogeneity on empirical results, in addition to adding multiple control variables of individual, family, and school types to control the influence of related factors, this study also adopted multiple robustness tests for testing to obtain reliable empirical results. Third, the research provides empirical support and new explanations for the application of relevant theories in the field of entrepreneurship. Using human capital theory, other theories, and empirical

evidence, this paper discusses the influence of EE on the EI of different students, deepens the understanding of the causes of the influence, and helps propose targeted policies and measures to promote entrepreneurship development. In addition, the literature on entrepreneurship is enriched by providing empirical evidence from China.

The remaining structure of this study is as follows: The second part proposes the research hypothesis through theoretical and literature analysis. The third part presents the research methods, including data, variables, empirical strategies, and model setting. The fourth part contains the empirical process and results, including baseline regression, robustness test, and heterogeneity analysis. The fifth part is the discussion, and the sixth part is the conclusion.

## Theoretical foundations and hypothesis development

### Theoretical foundations

Human capital theory regards the knowledge, skills, health, and values people possess as the capital embodied in them. Education and training are the most critical investments in human capital, influencing people's future monetary and spiritual earning activities by improving knowledge and skills [77–79]. From the human capital perspective, EE affects EI mainly by improving job opportunities, knowledge, and ability. Education enhances identifying and exploiting business opportunities and facilitates access to financial capital and material resources for entrepreneurship [80]. Education provides future entrepreneurs and the general helpful population with entrepreneurial skills through knowledge accumulation and promotion [81]. EE and training promote EIs for college students by improving knowledge, skills, and other abilities[15, 20, 39, 80, 82].

Self-efficacy theory refers to the influence of an individual's subjective evaluation of self-ability on behavioral motivation. Self-efficacy is self-knowledge of interest, motivation, perseverance, ability, and action that is positively correlated with EI and entrepreneurial behavior [1, 52, 83, 84]. EE is considered an educator's effort to positively influence students' entrepreneurial motivation and attitude [13, 58]. EE forms EIs by generating the motivation to perceive new opportunities and the adequacy of the means needed to meet entrepreneurial requirements [71, 85–88]. EE enables college students to enhance the perception of individual and collective entrepreneurial efficiency and feasibility, thus improving entrepreneurial attitude and EI [2, 17, 49, 59–65].

The theory of planned behavior combines the attitude of behavior, the feeling of social pressure, and the perception of behavioral control into individual intention [89, 90]. According to the theory of planned behavior, EE also improves college students' understanding and control of self-behavior and external pressure, thus enhancing their EI [12, 25, 34, 64, 66–72].

In summary, human capital theory from knowledge and skills, self-efficacy theory from the perspective of self-subjective evaluation, and planned behavior theory from the perspective of attitude, external pressure, and behavior control provide a theoretical basis for the impact of entrepreneurial education on EI. This study will take these theories as the framework and in-depth analysis of the relevant influencing reasons and mechanisms to provide theoretical support for studying EE on EI.

### The effect of EE on EI

EE refers to the educational process of cultivating the entrepreneurial spirit, knowledge, and skills; EI is a prerequisite for actual entrepreneurial behavior [42]. Scholars differ on the effect of EE on college students' EI.

Most scholars believe EE positively correlates with college students' EI [1, 19, 24, 25, 35, 38, 42–45]. The course teaching and practical activities of EE have significantly improved the

entrepreneurial knowledge, analytical ability, entrepreneurial attitude, and entrepreneurial willingness of college students [13, 34, 68, 69, 75, 91–93]. Entrepreneurship courses create a positive atmosphere and promote students' EIs [37, 75, 92]. According to human capital theory, EE positively impacts students' entrepreneurial activities by improving their knowledge and entrepreneurial skills [54, 55, 94]. Farashah (2013) found that the possibility of starting a business increases 1.3 times for each unit improvement in EE or one entrepreneurship course [56]. Barba-Sanchez and Atienza-Sahuquillo (2018) found that after EE and training, the EI of college students increased by approximately 30% [57]. From the perspective of self-efficacy theory, EE builds self-efficacy in the aspects of entrepreneurial attitude, entrepreneurial interest, and entrepreneurial ability to improve EI [95, 96]. EE can improve students' EI by cultivating entrepreneurial cognition, attitude, and spirit [2, 17, 57, 62, 64, 70, 72, 97]. EE improves students' EI by improving their entrepreneurial efficiency and perception of entrepreneurial feasibility [2, 49, 59–61, 63, 65, 66, 98]. Regarding attitude and social pressure, EE increases the risk-taking spirit of college students and reduces the fear of failure, and these factors have a significant positive correlation with EI [50, 99, 100].

However, other scholars believe that the impact of EE on EI is minor or even negative [22, 46]. Imperfect EE does not improve the human capital level of college students, thus hindering the development of students' entrepreneurial desires [23, 101]. EE's curriculum and training still have problems cultivating students' creativity [67, 76, 102]. EE makes students understand the difficulty of entrepreneurship, which reduces their EI [47]. EE does not affect some students or even has a reverse effect. The reason is that EE makes students without entrepreneurial spirit realize that they are not suitable to engage in entrepreneurial activities but reduces their EI [48]. EE raises the EIs of students with no entrepreneurial experience but lowers the EIs of students with initial entrepreneurial experience [49].

In summary, EE can improve students' EI by improving human capital, self-efficacy, and control ability. Compared with the literature, this study adds control variables that affect individual ability and entrepreneurial experience, such as academic performance, class cadre, internships, part-time jobs, etc. It carries out robustness analysis to help obtain accurate and reliable empirical results. China is divided into eastern, central, and western regions in terms of economic development from high to low. There have been some studies on EE's effect on college students' EI in China, mainly in the economically developed east [2, 25, 53] and the central region [68], while there are few studies on the western region. The region of this study belongs to the economically underdeveloped western region to increase the relevant empirical evidence from the relatively underdeveloped regions of China.

It should be pointed out that a few kinds of literature on the influence of EE on EI are slight or even negative, mainly due to the following reasons. First, EE has a weak influence on students who have entrepreneurial experience, good family background and have received EE before entering school [22, 49, 101]. Second, endogeneity may lead to reverse effects of entrepreneurial education on EI [46, 48]. Third, economic and political instability may cause people to seek stable jobs rather than start their businesses, thus making EE have an inverse relationship with EI [23]. The following factors make it difficult for this study to show the reverse result of EE on EI. First, the research samples are from full-time college students in relatively underdeveloped areas of China, who generally have no entrepreneurial experience and ordinary family status. Second, the empirical study has addressed the endogeneity problem by adding control variables and robustness tests. Third, China's economy and society have enjoyed sustained and stable development. Accordingly, hypothesis 1 is proposed in this study:

H1. EE has a positive impact on college students' EIs.

## EE on EI: Gender differences

There are gender differences in the influence of EE on EI [35, 60, 68]. EE is more conducive to improving women's human capital level that that of men, thus increasing the probability of entrepreneurship participation [42]. EE improves women's entrepreneurial knowledge and skills and strengthens women's confidence to participate in entrepreneurship [99]. Although men's perception of the entrepreneurial environment is more vital than women's [70, 75], EE has a more significant effect on improving female self-efficacy, enabling women to overcome the limitation of low self-efficacy and increasing the probability of female entrepreneurial participation and success [52, 103]. EE has a more significant effect on the improvement of female self-efficacy than that of males. Since self-efficacy positively affects EI, the improvement of female EI is more significant than male EI [63, 104]. There are significant differences in risk tolerance and behavioral control between men and women [35, 39, 53, 96, 105, 106]. Without receiving EE, male college students have more EI than female students [50] because male students are better than female students in innovation ability and entrepreneurial control [44, 95]. After receiving EE, women increase their attitude and control their ability to take risks so that the effect of improving EI is better than that of men.

Scholars have different views on the gender differences in EE and EI. The first view holds that the effect of entrepreneurial education on EI is higher for women than for men because entrepreneurial education improves women's self-efficacy [22] and provides more entrepreneurial choices for women [51]. The second view holds that there are no gender differences in the impact of EE on college students' EI, and gender does not regulate the relationship between EE and EI [107]. The third view holds that the effect of EE on EI is higher in men than in women. Walter et al. (2013) found that entrepreneurial education and EI were positively correlated only among men but not among women because women needed to balance work and family, and there was a lack of employment promotion strategies for women [39]. Zhang et al. (2014) conducted a study on 10 universities in China and found that after receiving EE, the probability of starting a business for men is 80% higher than that for women [53]. Then, after receiving EE, does the EI of female and male students differ significantly due to self-efficacy and other reasons? To answer this question, the following research hypothesis 2a is proposed:

H2a. The influence of EE on the EI of male and female college students is significantly different.

## EE on EI: Differences between urban and rural college students

Most studies of EIs ignore regional differences [74]. Moreover, studies on the rural–urban differences in EIs are inconsistent. One view is that college students in urban environments have richer entrepreneurial resources and conditions, so their EI and success rate are higher than those in rural environments [44]. However, another view is that urban students are more likely to work in family businesses and obtain stable jobs, thus reducing their EI. The rural environment increased the probability of participation in entrepreneurship but did not affect entrepreneurial success. Family enterprises may reduce urban college students' EI, but it is easier for them to succeed because they can obtain the support of family resources [35]. In China, urban college students generally have a higher level of knowledge and skills in entrepreneurship than rural students. After receiving EE, is there a difference between urban and rural students regarding human capital, self-efficacy, entrepreneurial attitude, social pressure, behavioral control, and other aspects, thus having a different impact on EI? To evaluate whether the EI of urban students or rural students is more significant after receiving EE, this study proposes research hypothesis 2b:

H2b. The impact of EE on the EI of urban and rural college students is significantly different.

## The impact of EE on EI: Differences between public and private universities

There are few types of research on the impact of EE in different types of universities on EI, and the empirical results are inconsistent. Some scholars believe that the impact of EE on EI is greater for college students in public universities than for those in private universities. For example, the stronger the research-oriented attributes of German public university departments are, the higher the EI of students is [39]. Other scholars believe that after students in private universities receive EE, the effect is more significant than that of students in public universities. Bergmann et al. (2018), based on a study of 8009 college students from 22 public universities in Germany, found that compared with applied universities, in public universities, insufficient support for students' entrepreneurial activities and a poor entrepreneurial atmosphere was linked to a negative impact on students' EI [75]. Zhang et al. (2014) used the probit model to study 494 college students from 10 universities in China and found that after receiving EE, the probability of EI of students in applied universities is 50% greater than that of students in research universities [53]. In China, public universities attach more importance to innovation due to their vital research attributes, while private universities attach more importance to entrepreneurship due to their vital application attributes. Public universities focus on theoretical teaching, mainly training innovative talent in research, design, and technology management and focusing on research results such as projects, patents, and papers. Private universities focus on application and practice; talent training is more closely connected with employment positions. After receiving EE, do students from public universities promoting research talent or those from private universities promoting applied talent have more EI? To answer this question, hypothesis 2c was proposed in this study:

H2c. The impact of EE on students' EI in public and private universities is significantly different.

## The impact of EE on EI: Differences between poor and nonpoor college students

Many Chinese college students are in poverty [4, 108]. EE helps poor college students participate in entrepreneurship and eradicate poverty [6, 13–17]. Poverty affects the EI of students. Nonpoor college students can obtain funds and social resources from their families to start their businesses [16]. Moreover, because nonpoor students do not need to make money immediately after graduation to support themselves and their families, they have more freedom to start their businesses. In contrast, it is difficult for poor college students to obtain the start-up capital and social capital needed for entrepreneurship from their families [109]. In addition, due to the economic difficulties of their families, they need to make money to support their families immediately after graduation, so they prefer to find stable nonentrepreneurial jobs, such as employment in large enterprises or government departments to obtain a stable salary, and their EI is relatively low. After receiving EE, does it increase the self-efficacy and entrepreneurial attitude of poor students, reduce the social pressure of poor students, and thus reduce the EI gap with nonpoor students? Alternatively, does EE increase the gap between the EI of poor and nonpoor students, leading to the phenomenon of an intergenerational poverty cycle in the entrepreneurial field? Hypothesis 2d is proposed in this study to evaluate the impact of EE on the EI of college students with different economic conditions:

H2d. There are significant differences in the impact of EE on the EI of poor and nonpoor college students.

## Methodology

### Ethics statement

This study was conducted under a research permit (No. EM-2022-A-15) from the School of Economics and Management, Guangxi University of Science and Technology, PRC. All participants signed a written informed consent form before data collection. Participants' information was strictly anonymous and confidential. The participants, all college students over 18 years old, had been informed of the purpose and use of the program and had the right to participate voluntarily and withdraw without reason.

### Data collection procedure and participants

Referring to the literature that selects students from one or two universities as samples [23, 38, 43, 44, 55, 57, 65, 68, 69, 110], this study selected students from two undergraduate universities in the Guangxi Zhuang Autonomous Region of China for a questionnaire survey in 2022. Class-based cluster sampling was used for the questionnaire, and paper questionnaires were distributed and collected on-site in the classroom under the supervision of teachers. A total of 569 questionnaires were returned. After collecting the questionnaire, according to the questions in the questionnaire, the data were entered into an Excel form. Then, the data in Excel form were converted into a format that can be recognized by Stata measurement software, and the data were coded. Omitted options in the questionnaire were treated as missing values. After deleting the samples with missing values, 518 questionnaires were obtained for the study, accounting for 91.04% of all questionnaires. The data covered the variables used in this study. Table 1 shows the specific variable definition and coding.

### Variable

The variables involved in this paper mainly include the dependent variable, independent variable, and control variable. The control variables include three categories: individual demographics, family status, and school status.

**Dependent variable.** The dependent variable is the students' EI. This variable definition is consistent with the literature that uses the dependent variable as EI and is a binary variable [37, 50, 52, 53, 56, 60, 98]. The variable comes from the question: If you have the opportunity and resources, are you likely to start a business? Students defined EI by choosing "1 = start a business, 0 = will not start a business" and formed a binary variable of EI.

**Independent variable.** The independent variable is EE. The questionnaire comes from the question: Do you agree that EE in school has increased your entrepreneurship knowledge and ability? The categorical variables of EE were formed from students' recognition of EE: 1 = strongly disagree, 2 = somewhat disagree, 3 = moderate, 4 = somewhat agree, and 5 = strongly agree.

**Individual control variable (individual).** Individual demographic control variables include six variables: age, gender, ethnic group, household registration, area of residence, and political status, which are used to control the impact of individual characteristics on EI. The reasons for choosing these control variables are as follows: first, EI changes with the age of students [35]. Second, men and women have different EI [34, 35, 37, 50, 60, 63, 68, 75, 95]. Third, ethnicity is related to EI [44, 95, 111]. The Han nationality is the largest ethnic group in China, and other ethnic groups are called ethnic minorities. In China, different employment

**Table 1. Definition of variables.**

| Variable | Definition |
|---|---|
| Entrepreneurship intention | Whether you are likely to start a business (0 = no, 1 = yes) |
| education | Whether the school's EE has increased your knowledge and ability of entrepreneurship (1 = strongly disagree, 2 = more disagree, 3 = medium, 4 = more agree, 5 = strongly agree) |
| age | Age range from 19 to 27 |
| gender | Gender (0 = female, 1 = male) |
| ethnicity | Ethnicity (0 = ethnic minorities, 1 = Han nationality) |
| hukou | Household registration (0 = urban, 1 = rural) |
| region | Area of residence (1 = western region, 2 = central Region, 3 = eastern region) |
| party | Whether you intend to join the party (0 = no, 1 = yes) |
| family_economic | The level of the economic status of the family in the location (1 = well below average, 2 = below average, 3 = average, 4 = above average, 5 = well above average) |
| siblings | Number of siblings (1 = zero, 2 = one, 3 = two, 4 = three, 5 = four or more) |
| father_edu_year | Years of education of the father |
| father_employ | Whether the father has his own business (0 = no, 1 = yes) |
| mother_employ | Whether the mother has her own business (0 = no, 1 = yes) |
| father_enterprise | Do your parents want you to start your own business? (1 = very much not hopeful, 2 = some do not hopeful, 3 = medium, 4 = Some hopeful, 5 = very hopeful) |
| major_type | Types of personnel training (1 = general undergraduate, 2 = vocational bachelor's degree, 3 = junior college starting point to undergraduate degree) |
| grade | Grade level (1 = freshman, 2 = sophomore, 3 = junior, 4 = senior) |
| class | There are 16 classes in total. |
| learn | Academic achievement grade (1 = very bad, 2 = a little bad, 3 = medium, 4 = good, 5 = excellent) |
| student_leader | Served as a student cadre (0 = no, 1 = yes) |
| student_club | Joined a student club (0 = no, 1 = yes) |
| volunteer | Participated in a student volunteer organization (0 = no, 1 = yes) |
| parttime_job | Part-time work experience (0 = no, 1 = yes) |
| internship | Internship experience (0 = no, 1 = yes) |

assistance policies for ethnic minorities may lead to different EIs. Fourth, household registration controls the difference in EI between urban and rural students caused by urban–rural differences. Fifth, according to the degree of economic development from low to high, China's regions are divided into the western, central, and eastern regions. Regional economic conditions affect EI [74, 112]. Sixth, political status controls the influence of ideological status on EI.

**Family control variable (family).** The family status variables include six variables: family economic status, the number of siblings, the father's education level, the father's and mother's entrepreneurial status, and the parents' entrepreneurial expectations for their children. The reason for choosing these control variables is that families influence the EI of their children [22, 42, 44, 74, 99]. First, starting a business needs financial support, and the family's economic situation is the basis for children to start a business. Second, the number of siblings causes the dilution of economic support and social resources, which affects the family's support for entrepreneurship. Third, the father's education level can provide better academic support for students to start a business, and the entrepreneurial experience of the father and mother affects their children's entrepreneurship through the role model effect [23, 31, 34, 35, 37, 49, 50, 60, 75].

**School control variable (school).** There are 10 variables related to school status: major, type of talent training, grade, class, academic performance, student cadre, student associations,

volunteerism, part-time work experience, and internships, to control for the differentiated impact of students' learning and environment in school on employment intention. The reasons for choosing these control variables are as follows: first, there are differences in EI among majors, types of talent training, grades, and classes in EE [44, 50, 53, 60, 67, 70, 95, 102]. Among them, talent training type refers to the different enrollment modes and training focuses; talent training is divided into three categories: ordinary undergraduate programs, vocational undergraduate programs, and junior college programs. General undergraduate refers to the traditional four-year undergraduate talent training method based on theoretical teaching. Vocational undergraduate refers to the four-year undergraduate talent training mode closely connected with employment positions, which attaches importance to practical operation. The junior college starting point for a bachelor's degree refers to a two-year undergraduate degree in a bachelor's degree university after completing a three-year college or university, called a "3+2" junior college starting point for a bachelor's degree in China. Second, student achievement levels are related to students' knowledge and ability, which affects EI [44, 113]. Third, whether students serve as student cadres, participate in student associations, and volunteer makes a difference in students' communication ability and affects their EI. Fourth, students' part-time jobs and internship experience in enterprises enable students to have prior entrepreneurial experience, knowledge, and ability, which affects EI [49, 95].

## Statistical description

Table 2 reports the descriptive statistics of the variables. For subsequent analysis, the following is the statistical distribution of some variables. Regarding the distribution of EI, those with EI are slightly lower than the average: 240 (46.33%) with EI and 278 (53.67%) without EI. The medium and medium-high levels of EE accounted for a large proportion: 1 = strongly disagree, 17 (3.28%); 2 = disagree, 48 (9.27%); 3 = medium, 267 (51.54%); 4 = agree, 170 (32.82%); and 5 = strongly agree, 16 (3.09%). Regarding gender, due to the analysis of engineering majors, there is a large proportion of males: 380 (73.36%) for males and 138 (26.64%) for females. In terms of ethnic distribution, as it is an ethnic minority autonomous region, the proportion of students from ethnic minorities is relatively high: 153 (29.54%) who are ethnic minorities and 365 (70.46%) who have Han nationality. Regarding household registration, as the university is in the economically underdeveloped western region of China and is mainly composed of local students, the proportion of rural college students is enormous: 100 (19.31%) urban college students and 418 (80.69%) rural college students. In terms of talent training types, 211 (40.73%) were general undergraduate students, 211 (40.73%) were vocational undergraduate students, and 96 (18.53%) had upgraded from a junior college to a bachelor's degree program. In terms of the distribution of academic performance, the grades are mainly medium to good: 1 = very bad, 13 (2.5%); 2 = a little bad, 65 (12.55%); 3 = medium, 278 (53.67%); 4 = good, 140 (27.03%); 5 = excellent, 22 (4.25%). The overall alpha value of the questionnaire is 0.7166, indicating good reliability.

## Empirical strategy and models

This study uses the econometric empirical method to verify the five hypotheses proposed. The following identification strategies are adopted to avoid the impact of endogeneity problems such as omitted variables on the empirical results. First, as many control variables as possible are added. To reduce the endogeneity problem caused by omitted variables, control variables such as demography, family status, and school status are added to control the relevant influencing factors as much as possible. Second, the robustness test is carried out. The method of replacing independent variables is adopted to reduce the influence caused by mutual causality.

**Table 2. Statistical description of variables.**

| Variable | Obs | Mean | SD | Min | Median | Max |
|---|---|---|---|---|---|---|
| Entrepreneurship intention | 518 | 0.4633 | 0.4991 | 0 | 0 | 1 |
| education | 518 | 3.2317 | 0.7894 | 1 | 3 | 5 |
| age | 518 | 21.7220 | 1.5374 | 19 | 22 | 27 |
| gender | 518 | 0.7336 | 0.4425 | 0 | 1 | 1 |
| ethnicity | 518 | 0.7046 | 0.4566 | 0 | 1 | 1 |
| hukou | 518 | 0.8069 | 0.3951 | 0 | 1 | 1 |
| region | 518 | 1.3436 | 0.6506 | 1 | 1 | 3 |
| party | 518 | 0.3977 | 0.4899 | 0 | 0 | 1 |
| family_economic | 518 | 2.5347 | 0.6854 | 1 | 3 | 5 |
| siblings | 518 | 2.4595 | 1.0850 | 1 | 2 | 5 |
| father_edu_year | 518 | 9.2934 | 2.6342 | 0 | 9 | 19 |
| father_employ | 518 | 0.1795 | 0.3842 | 0 | 0 | 1 |
| mother_employ | 518 | 0.1429 | 0.3503 | 0 | 0 | 1 |
| father_enterprise | 518 | 2.9788 | 0.8142 | 1 | 3 | 5 |
| major | 518 | 1.7181 | 1.1408 | 1 | 1 | 4 |
| major_type | 518 | 1.7780 | 0.7379 | 1 | 2 | 3 |
| grade | 518 | 2.7432 | 0.9400 | 1 | 3 | 4 |
| class | 518 | 9.1274 | 4.7939 | 1 | 10 | 16 |
| learn | 518 | 3.1795 | 0.7969 | 1 | 3 | 5 |
| student_leader | 518 | 0.5347 | 0.4993 | 0 | 1 | 1 |
| student_club | 518 | 0.7819 | 0.4134 | 0 | 1 | 1 |
| volunteer | 518 | 0.7529 | 0.4317 | 0 | 1 | 1 |
| parttime_job | 518 | 0.6757 | 0.4686 | 0 | 1 | 1 |
| internship | 518 | 0.4768 | 0.4999 | 0 | 0 | 1 |

The method of changing the sample range of data is adopted to reduce the problem of measurement bias and sample self-selection. Linear probability and probit models are used for regression to reduce the impact of model setting errors. Third, heterogeneity analysis is performed. To avoid the problem of inconsistent group data after sample grouping, the method of including interaction terms in the regression is not adopted, but the method of grouping regression is used for heterogeneity analysis. Through the above methods, the effect and difference of EE on EI are quantitatively evaluated from the numerical magnitude and statistical significance indicators of regression results.

Since the dependent variable is binary, the study uses a logistic model. The independent variable is EE ($x$), and the dependent variable is EI ($Y$). Assuming that the probability of having an EI is $P$ and the probability of not having an EI is $(1-P)$, the regression model is Model (1):

$$\ln\left(\frac{P}{(1-P)}\right) = \alpha + \beta_1 x_1 + \cdots + \beta_n x_n \tag{1}$$

The formula $x_1, x_2, \cdots, x_n$ represents the factors that impact the dependent variable; $\alpha$ is a constant term $\beta_1, \beta_2, \cdots, \beta_n$ and is the partial regression coefficient of logistic regression. By changing Eq (1), the odds ratio model (2) can be obtained:

$$odds = \frac{P}{(1-P)} = \exp(\alpha + \beta_1 x_1 + \cdots + \beta_n x_n) \tag{2}$$

The logistic model is used in all other regressions except for the regression for the robustness test of changing the empirical model. In the logistic model, the regression coefficients can only

judge the direction of influence according to the positive and negative signs but cannot be compared numerically according to the coefficient size. For better quantitative comparison, the regression output result is set as the odds ratio (OR), where a number greater than 1 represents a positive impact, a number less than 1 represents a negative impact, and the regression coefficient minus 1 represents the corresponding increase or decrease in probability. For example, the odds ratio is 1.7249, indicating that for each grade of EE, the probability of a student's EI increases by 72.49%. In the table, each column represents a regression model. To simplify the table, the regression results of all control variables are output in the benchmark regression table, and only the regression results of independent variables on dependent variables are output in the robustness test and heterogeneity analysis. The three control variables, individual, family, and school, are denoted by "Yes" to indicate control, and a blank indicates that such control variables are not controlled for. In selecting control variables, all control variables are included in the benchmark regression and robustness test. All control variables are included in the regressions except for a slightly adjusted control variable for the heterogeneity analysis of school type and poverty status.

## Results

### Benchmark regression: The impact of EE on the EI of college students

Table 3 reports the empirical results of the benchmark regression of EE on the EI of college students. Column (1) shows the regression results without adding any control variables. For each additional level of EE, the probability of starting a business increases by 112.17% (OR = 2.1217, $P < 0.01$). Column (2) controls for individual characteristics. For each additional level of EE, the probability of starting a business increases by 102.46% (OR = 2.0246, $P < 0.01$). Column (3) controls for individual and family variables. For each additional level of EE, the probability of starting a business increases by 99.51% (OR = 1.9951, $P < 0.01$). Column (4) includes all the control variables. For each additional level of EE, the probability of starting a business increases by 86.34% (OR = 1.8634, $P < 0.01$). After gradually adding control variables, the positive effect of EE on EI gradually decreases. This shows that individual, family, school, and other factors affect the empirical results, and the empirical results without control variables overestimate the impact of EE on EI. The empirical results in Table 3 support H1 that EE significantly impacts students' EI.

### Robustness test

Endogeneity problems include missing variables, reverse causality, sample selection bias, and measurement bias between EE and EI. For example, there are unobserved individual differences, such as students' abilities and psychological characteristics [35, 36, 42, 46]. To reduce the impact of endogeneity on empirical results, in addition to using control variables to control relevant influencing factors, robustness tests such as changing the core explanatory variable index, changing the sample scope, and changing the model are also adopted to effectively deal with the endogeneity problem and ensure the robustness of empirical results.

**Replacing the independent variable.** There is an endogeneity problem of reverse causality between EE and EI. Students' recognition of the educational effect of EE will affect their EI, and EI will also affect students' enthusiasm to participate in EE and their recognition of the educational effect. To this end, this study uses the method of replacing independent variables for the empirical test. Specifically, this study uses students' judgment on the necessity of EE as a surrogate variable for the original independent variable. Since the entrepreneurship course is compulsory for every student, it is not directly related to whether students have EI and participate in entrepreneurial activities and competitions, thus reducing the impact of EE on the reverse causality of EI. The necessity of EE (variable name: edu_necessary) comes from the

**Table 3. The impact of EE on the EI of college students: Benchmark regression.**

| Variable | Dependent variable: Entrepreneurship intention | | | |
|---|---|---|---|---|
| | (1) | (2) | (3) | (4) |
| education | 2.1217*** | 2.0246*** | 1.9951*** | 1.8634*** |
| | (0.2739) | (0.2631) | (0.2641) | (0.2495) |
| age | | 1.0037 | 0.9943 | 1.0006 |
| | | (0.0618) | (0.0637) | (0.1089) |
| gender | | 0.9165 | 0.7862 | 0.7455 |
| | | (0.1984) | (0.1765) | (0.1797) |
| ethnicity | | 0.8775 | 0.8569 | 0.8488 |
| | | (0.1824) | (0.1831) | (0.1882) |
| hukou | | 1.2824 | 1.1338 | 1.0762 |
| | | (0.3094) | (0.3051) | (0.2972) |
| region | | 0.6988** | 0.7254* | 0.8445 |
| | | (0.1104) | (0.1208) | (0.1580) |
| party | | 0.8181 | 0.8220 | 0.7582 |
| | | (0.1601) | (0.1661) | (0.1634) |
| family_economic | | | 0.9744 | 0.9918 |
| | | | (0.1460) | (0.1516) |
| siblings | | | 0.8892 | 0.8885 |
| | | | (0.0844) | (0.0863) |
| father_edu_year | | | 0.9540 | 0.9574 |
| | | | (0.0386) | (0.0397) |
| father_employ | | | 0.9371 | 0.9080 |
| | | | (0.3007) | (0.3016) |
| mother_employ | | | 0.8987 | 0.9331 |
| | | | (0.3065) | (0.3330) |
| father_enterprise | | | 1.7998*** | 1.7233*** |
| | | | (0.2317) | (0.2259) |
| major | | | | 1.1627 |
| | | | | (0.3537) |
| major_type | | | | 0.7045 |
| | | | | (0.1803) |
| grade | | | | 0.6029 |
| | | | | (0.2067) |
| class | | | | 0.8343* |
| | | | | (0.0854) |
| learn | | | | 1.0345 |
| | | | | (0.1347) |
| student_leader | | | | 1.2677 |
| | | | | (0.2737) |
| student_club | | | | 1.6387* |
| | | | | (0.4263) |
| volunteer | | | | 0.8278 |
| | | | | (0.2139) |
| parttime_job | | | | 1.1077 |
| | | | | (0.2508) |
| internship | | | | 0.9972 |
| | | | | (0.2407) |

(*Continued*)

**Table 3.** (Continued)

| Variable | Dependent variable: Entrepreneurship intention | | | |
|---|---|---|---|---|
| | **(1)** | **(2)** | **(3)** | **(4)** |
| **Constant** | 0.0746*** | 0.1334 | 0.0806 | 1.4688 |
| | (0.0323) | (0.1971) | (0.1333) | (4.1869) |
| **Observations** | 518 | 518 | 518 | 518 |
| **Pseudo R-squared** | 0.0544 | 0.0691 | 0.1030 | 0.1260 |

Note: Regression values are odds ratios. The values in parentheses are standard errors.

*** p<0.01

** p<0.05

* p<0.1

question, "Do you think schools must carry out EE?". Students responded by selecting "1 = very unnecessary, 2 = relatively unnecessary, 3 = moderate, 4 = relatively necessary, 5 = very necessary" and assigned values of 1–5 in turn to form the replacement variable of the original independent variable.

Table 4 reports the robustness test results after replacing the independent variables. Column (1) shows the results without adding any control variables. For each additional level of EE, the probability of starting a business increases by 70.31% (OR = 1.7031, $P < 0.01$). Column (2) controls for individual variables. For each additional level of EE, the probability of starting a business increases by 67.43% (OR = 1.6743, $P < 0.01$). Column (3) controls for individual and family variables. For each additional level of EE, the probability of starting a business increases by 62.00% (OR = 1.6200, $P < 0.01$). Column (4) includes all the control variables. For each additional level of EE, the probability of starting a business increases by 47.91% (OR = 1.4791, $P < 0.01$). The results in Table 4 show that the regression results after replacing the independent variables all show a positive and significant relationship, indicating that the benchmark regression results are robust. The empirical results of the robustness test support H1, indicating that EE has a significant positive impact on EI.

**Deleting the sample of undergraduate students from junior college.** The study suffers from the endogeneity problem of sample selection bias. The relationship between EE and EI is strongly influenced by individuals who have already had EE and entrepreneurial experience before receiving EE [31, 49, 68, 95]. Students who have taken entrepreneurship courses or have entrepreneurial experience may have EI before receiving EE in college, so the impact of EE on EI is not the actual result of EE [43]. To reduce the sample selection bias, this study uses the deletion of the samples of students who have graduated from junior college to enter a bachelor's degree program to overcome the influence of learning and entrepreneurial experience before receiving undergraduate EE and conduct a robustness test. Students from the junior college starting point for an undergraduate degree had EE and practice experience before enrolling in an undergraduate university. Students had taken entrepreneurial courses and practical training during their three-year junior college study and generally had more than three months of paid work experience in enterprises or society. After working in enterprises or starting their own business for several years after junior college graduation, some students already had practical entrepreneurial and working experience and returned to study in two-year undergraduate universities.

Table 5 reports the empirical results of the robustness test of deleting the samples of students who graduated from junior college with bachelor's degrees. Column (1) shows the results without adding any control variables. For each additional level of EE, the probability of starting

**Table 4. Replacing the independent variable: Robustness test.**

| Variable | Dependent variable: Entrepreneurship intention | | | |
|---|---|---|---|---|
| | (1) | (2) | (3) | (4) |
| **edu_necessary** | 1.7031*** | 1.6743*** | 1.6200*** | 1.4791*** |
| | (0.1831) | (0.1829) | (0.1796) | (0.1697) |
| **Individual** | | Yes | Yes | Yes |
| **Family** | | | Yes | Yes |
| **School** | | | | Yes |
| **Constant** | 0.1219*** | 0.2788 | 0.1756 | 2.0123 |
| | (0.0497) | (0.4022) | (0.2832) | (5.6707) |
| **Observations** | 518 | 518 | 518 | 518 |
| **Pseudo R-squared** | 0.0375 | 0.0564 | 0.0892 | 0.1110 |

Note: Regression values are odds ratios. The values in parentheses are standard errors.

*** p<0.01

** p<0.05

* p<0.1

a business increases by 140.60% (OR = 2.4060, $P < 0.01$). Column (2) controls individual variables. For each additional level of EE, the probability of starting a business increases by 125.53% (OR = 2.2553, $P < 0.01$). Column (3) controls individual and family variables. For each additional level of EE, the probability of starting a business increases by 120.23% (OR = 2.2023, $P < 0.01$). Column (4) includes all the control variables. For each additional level of EE, the probability of starting a business increases by 109.57% (OR = 2.0957, $P < 0.01$). After deleting the samples of students with EE and entrepreneurial experience, the impact of EE on EI increases significantly compared with the value in the benchmark regression (see Table 1). This shows that the sample of students who already have EE and experience significantly reduces the impact of EE on EI, and the impact of EE on the EI of these students is small. After deleting the samples of students with EE and experience, the regression results all show a positive and significant relationship, indicating that the benchmark regression results are robust. The empirical results support H1, indicating that EE significantly positively affects EI.

**Deleting the samples with the best and worst academic performance.** The study suffers from the endogeneity problems of sample self-selection and measurement bias. First, academic performance is related to students' knowledge and ability, and the learning attitude, knowledge, and skills of excellent and poor students in EE are inconsistent, thus affecting the effect of EI [113]. Second, although every student must take the entrepreneurship course, students with excellent performance may take the initiative to participate in entrepreneurial activities and competitions, which leads to the nonrandom selection of samples and the self-selection bias of samples. Third, in terms of endogeneity problems caused by measurement bias, students with poor performance do not show their true intentions when filling out the questionnaires due to reasons such as academic burden, lack of interest in learning, and lack of entrepreneurship, so they fill in the questionnaires at will, resulting in measurement bias. To reduce the endogeneity problem of sample self-selection and measurement bias, this paper removes the samples of students with the best and worst academic performance grades and reruns the regression for the robustness test.

Table 6 reports the results of the robustness test after deleting the sample. Column (1) does not include any control variables. For each additional level of EE, the probability of starting a

**Table 5. Deleting the sample of undergraduate students from junior college: Robustness test.**

| Variable | Dependent variable: Entrepreneurship intention | | | |
|---|---|---|---|---|
| | **(1)** | **(2)** | **(3)** | **(4)** |
| **education** | 2.4060*** | 2.2553*** | 2.2023*** | 2.0957*** |
| | (0.3579) | (0.3417) | (0.3405) | (0.3371) |
| **Individual** | | Yes | Yes | Yes |
| **Family** | | | Yes | Yes |
| **School** | | | | Yes |
| **Constant** | 0.0449*** | 5.507 | 2.8491 | 224.1388 |
| | (0.0226) | (11.4075) | (6.4837) | (800.3093) |
| **Observations** | 422 | 422 | 422 | 422 |
| **Pseudo R-squared** | 0.0714 | 0.0982 | 0.1250 | 0.1490 |

Note: Regression values are odds ratios. The values in parentheses are standard errors.

*** p<0.01

** p<0.05

* p<0.1

business increases by 139.23% (OR = 2.3923, $P < 0.01$). Column (2) controls individual variables. For each additional level of EE, the probability of starting a business increases by 125.12% (OR = 2.2512, $P < 0.01$). Column (3) controls individual and family variables. For each additional level of EE, the probability of starting a business increases by 119.38% (OR = 2.1938, $P < 0.01$). Column (4) includes all the control variables. For each additional level of EE, the probability of starting a business increases by 101.38% (OR = 2.0138, $P < 0.01$). After deleting the samples of students with the best and worst academic performance grades, the impact of EE on EI increases significantly compared with the value in the benchmark regression (see Table 1). This shows that the samples of students with the best and worst academic performance grades significantly reduce the impact of EE on EI, and the students with the worst academic performance have a more significant negative impact on the regression results. The results in Table 6 show that after deleting the samples of students with the best and worst academic performance grades, the regression results show a positive and significant relationship, indicating that the results of the benchmark regression are robust. The empirical results of the robustness test support H1, indicating that EE has a significant positive impact on EI.

**Changing the empirical model.** To prevent the problems caused by model setting errors, this study adopts a linear probability model (LPM) and probit model for the robustness test. Table 7 reports the empirical results of the robustness test of the EE that changes the model on the EI of college students. Columns (1) and (2) are the estimation results using the LPM. Column (1) shows the regression results of the LPM without the control variables. For each additional level of EE, the proportion of EI increases by 16.88% ($P < 0.01$). Column (2) shows the regression results of the LPM with all control variables included. For each additional level of EE, the proportion of EI increases by 12.89% ($P < 0.01$). Columns (3) and (4) are the estimation results of the probit model. Since the coefficient value output by the probit model can only represent the impact's direction but not the impact's value, this study uses the marginal effect value for analysis. Column (3) does not include the regression results of the probit model of the control variable. For each additional level of EE, the probability of EI increases by 17.45% ($P < 0.01$). Column (4) shows the regression results of the probit model with all control variables included. For each additional level of EE, the probability of EI increases by 14.78% ($P < 0.01$). The empirical results support H1. The robustness regression results after

**Table 6. Deleting the samples with the best and worst academic performance: Robustness test.**

| Variable | Dependent variable: Entrepreneurship intention | | | |
|---|---|---|---|---|
| | (1) | (2) | (3) | (4) |
| education | 2.3923*** | 2.2512*** | 2.1938*** | 2.0138*** |
| | (0.3414) | (0.3260) | (0.3269) | (0.3032) |
| Individual | | Yes | Yes | Yes |
| Family | | | Yes | Yes |
| School | | | | Yes |
| Constant | 0.0483*** | 0.1060 | 0.0430* | 0.2778 |
| | (0.0232) | (0.1675) | (0.0763) | (0.8566) |
| Observations | 483 | 483 | 483 | 483 |
| Pseudo R-squared | 0.0656 | 0.0845 | 0.1230 | 0.1430 |

Note: Regression values are odds ratios. The values in parentheses are standard errors. *** p<0.01

** p<0.05

* p<0.1

changing the model still show a significant positive relationship, indicating that the benchmark regression results are robust.

## Heterogeneity analysis

To analyze the impact of EE on different types of college students' EI, this study conducts heterogeneity analysis by grouping regression from four perspectives: male and female, urban and rural, public and private universities, and poor and nonpoor students.

**Gender differences in EE on EI.** Table 8 reports the difference in the impact of EE on the EI of male and female college students. The results in column (1) show that when no control variables are added, the possibility of entrepreneurship for men and women increases by 104.02% (OR = 2.0402, $P < 0.01$) and 134.13% (OR = 2.3413, $P < 0.01$) for each additional level of EE. This indicates a significant gender difference, and EE has a better effect on

**Table 7. Changing the empirical model: Robustness test.**

| Variable | Dependent variable: Entrepreneurship intention | | | |
|---|---|---|---|---|
| | LPM | | Probit | |
| | (1) | (2) | (3) | (4) |
| education | 0.1688*** | 0.1289*** | 0.1745*** | 0.1478*** |
| | (0.0268) | (0.0273) | (0.0290) | (0.0310) |
| Individual | | Yes | | Yes |
| Family | | Yes | | Yes |
| School | | Yes | | Yes |
| Constant | -0.0823 | 0.5471 | | |
| | (0.0892) | (0.6081) | | |
| Observations | 518 | 518 | 518 | 518 |
| R-squared | 0.0713 | 0.1590 | | |
| Pseudo R-squared | | | 0.0525 | 0.1270 |

Note: The regression values of the LPM are the coefficient values. The regression value of the Probit is the marginal effect. The values in parentheses are standard errors.

*** p<0.01

** p<0.05

* p<0.1

promoting women's EI. After adding the individual control variables, the results in column (2) show that the probability of starting a business increases by 99.81% (OR = 1.9981, $P < 0.01$) for men and 116.74% (OR = 2.1674, $P < 0.01$) for women when the education of entrepreneurship increases by one level. This indicates that the gender differences and the more significant effect on promoting female EI still exist after controlling for individual demographic characteristics. After controlling for individual and family factors, the results in column (3) show that the probability of starting a business increases by 100.39% (OR = 2.0039, $P < 0.01$) and 108.22% (OR = 2.0822, $P < 0.01$) for men and women, respectively, with each increase of one level of EE. After controlling for family factors, the differences between the two showed a decreasing trend, indicating that family background is an essential factor affecting gender differences. When all the control variables are included, the results in column (4) show that the gender difference is further expanded, with the possibility of starting a business for men and women increasing by 83.55% (OR = 1.8355, $P < 0.01$) and 131.92% (OR = 2.3192, $P < 0.01$), respectively, for each additional level of EE. This indicates that EE has a more significant effect on improving women's EI under the same learning and practice performance. The empirical results support hypothesis H2a. Whether the control variables are added or not does not change the result that there is a significant gender difference between EE and EI. Before receiving EE, women's EI was lower than men's EI [50]. However, EE can improve the EI of women more than that of men, narrowing the entrepreneurial gap between women and men.

**Differences in EE on EI between rural and urban college students.** Table 9 reports the effect of EE on the EI of rural and urban college students. This study uses household registration to define rural and urban college students: those with agricultural household registration are rural college students, and those with nonagricultural household registration are urban college students. Column (1) shows the results without adding control variables. With an increase of one level in EE, the possibility of entrepreneurship of rural and urban students increased by 79.94% (OR = 1.7994, $P < 0.01$) and 488.70% (OR = 5.8870, $P < 0.01$), respectively. This indicates a significant difference between urban and rural areas, and that EE has a better effect on improving the EI of urban college students. Column (2) shows the results of adding individual control variables. The EE increased by one level, and the possibility of starting a business increased by 73.71% (OR = 1.7371, $P < 0.01$) and 620.27% (OR = 7.2027, $P < 0.01$) for rural and urban students, respectively. EE further increases the EI gap between urban and rural students after controlling for individual population characteristics. Column (3) is the result of considering the control variables of individual and family factors. With an increase of one level of EE, the possibility of entrepreneurship of rural and urban students increases by 67.26% (OR = 1.6726, $P < 0.01$) and 930.25% (OR = 10.3025, $P < 0.01$), respectively. Column (4) shows the results when all control variables are included. When EE increases by one level, the possibility of entrepreneurship of rural and urban students increases by 55.39% (OR = 1.5539, $P < 0.01$) and 1106.59% (OR = 12.0659, $P < 0.01$), respectively. The difference test value of the intergroup coefficient is significant at 0.05, indicating that urban college students have more EI after receiving EE. The empirical results support hypothesis H2b, which indicates that EE significantly differs in its impact on the EI of rural and urban students. According to the regression results of gradually adding control variables, urban college students benefit far more from EE than rural college students. This shows that EE further enhances the advantages of urban students' original family economic and social capital, thus reducing the possibility of rural students' participation in entrepreneurship.

**Differences between public and private universities in the impact of EE on EI.** Table 10 reports the differences between public and private university students in the impact of EE on EI. Column (1) shows the results without adding control variables. With an increase in EE, the likelihood of starting a business increased by 54.32% (OR = 1.5432, $P < 0.01$) and 425.19%

**Table 8. Gender differences.**

| Variable | Dependent variable: Entrepreneurship intention | | | |
|---|---|---|---|---|
| | **(1)** | **(2)** | **(3)** | **(4)** |
| **Panel A: Male** | | | | |
| education | 2.0402*** | 1.9981*** | 2.0039*** | 1.8355*** |
| | (0.3051) | (0.3004) | (0.3124) | (0.2949) |
| Individual | | Yes | Yes | Yes |
| Family | | | Yes | Yes |
| School | | | | Yes |
| Constant | 0.0826*** | 0.0946 | 0.0556 | 0.8506 |
| | (0.0412) | (0.1593) | (0.1055) | (2.7850) |
| Observations | 380 | 380 | 380 | 380 |
| Pseudo R-squared | 0.0491 | 0.0673 | 0.1110 | 0.1400 |
| **Panel B: Female** | | | | |
| education | 2.3413*** | 2.1674*** | 2.0822*** | 2.3192*** |
| | (0.6029) | (0.5720) | (0.5561) | (0.6578) |
| Individual | | Yes | Yes | Yes |
| Family | | | Yes | Yes |
| School | | | | Yes |
| Constant | 0.0576*** | 0.6924 | 0.2061 | 1.4037 |
| | (0.0506) | (2.1159) | (0.7360) | (9.0559) |
| Observations | 138 | 138 | 138 | 138 |
| Pseudo R-squared | 0.0680 | 0.0882 | 0.1060 | 0.1480 |
| Group difference | 0.1377 | 0.0814 | 0.0383 | 0.2339 |

Note: Regression values are odds ratios. The values in parentheses are standard errors. Differences in coefficients between groups were tested using Fisher's bootstrap based (1000 times) approach.

*** p<0.01

** p<0.05

* p<0.1

(OR = 5.2519, $P < 0.01$), respectively, among students from public and private universities. This indicates a significant difference in school types, and EE has a better effect on promoting EI in private universities. Column (2) shows the results of adding individual control variables. With an increase in the level of EE, the likelihood of starting a business increased by 54.77% (OR = 1.5477, $P < 0.01$) for students from public universities and 425.84% (OR = 5.2584, $P < 0.01$) for students from private universities. This indicates that EE further increases the EI gap between students in public and private universities after controlling for individual population characteristics. Column (3) is the result that includes the control variables of individual and family factors. With an increase of one level of EE, the possibility of entrepreneurship of students from public universities and private universities increases by 57.09% (OR = 1.5709, $P < 0.01$) and 409.57% (OR = 5.0957, $P < 0.01$), respectively. Column (4) shows the results when all control variables are included. When EE increases by one level, the entrepreneurial possibility of public and private universities increases by 47.44% (OR = 1.4744, $P < 0.01$) and 578.19% (OR = 6.7819, $P < 0.01$), respectively. The difference test value of the intergroup coefficient is significant at the level of 0.01, indicating that students in private universities have more EI after receiving EE. The empirical results support hypothesis H2c, that EE has a significant difference in the impact of EI on students from public universities and private universities. According to the regression results, after receiving EE, the effect of enhancing the EI of

**Table 9. Differences between urban and rural college students.**

| Variable | Dependent variable: Entrepreneurship intention | | | |
|---|---|---|---|---|
| | (1) | (2) | (3) | (4) |
| **Panel A: Rural** | | | | |
| education | 1.7994*** | 1.7371*** | 1.6726*** | 1.5539*** |
| | (0.2446) | (0.2382) | (0.2360) | (0.2255) |
| Individual | | Yes | Yes | Yes |
| Family | | | Yes | Yes |
| School | | | | Yes |
| Constant | 0.1364*** | 0.2704 | 0.0984 | 1.8078 |
| | (0.0623) | (0.4283) | (0.1735) | (5.6205) |
| Observations | 418 | 418 | 418 | 418 |
| Pseudo R-squared | 0.0354 | 0.0454 | 0.0831 | 0.1160 |
| **Panel B: Urban** | | | | |
| education | 5.8870*** | 7.2027*** | 10.3025*** | 12.0659*** |
| | (2.4457) | (3.3875) | (5.6590) | (8.1387) |
| Individual | | Yes | Yes | Yes |
| Family | | | Yes | Yes |
| School | | | | Yes |
| Constant | 0.0020*** | 0.0036 | 0.0344 | 0.0025 |
| | (0.0028) | (0.0162) | (0.1813) | (0.0245) |
| Observations | 100 | 100 | 100 | 100 |
| Pseudo R-squared | 0.1990 | 0.2440 | 0.3250 | 0.3960 |
| Group difference | 1.1853** | 1.4222*** | 1.8180*** | 2.0496 *** |

Note: Regression values are odds ratios. The values in parentheses are standard errors. Differences in coefficients between groups were tested using Fisher's bootstrap based (1000 times) approach.

*** p<0.01

** p<0.05

* p<0.1

students in private universities is higher than that of students in public universities. This indicates that EE has a more significant effect on the EI of students from private universities focusing on application-oriented talent training than students from public universities focusing on research-oriented talent training.

**Differences in EE on EI between poor and nonpoor college students.** Table 11 reports the effects of EE on college students' EI in poor and nonpoor college students. Regarding poverty status, the poor students identified by the school are used as the standard to distinguish whether they are poor. Column (1) shows the results without adding control variables. When EE increases by one level, the possibility of entrepreneurship of poor students and nonpoor students increases by 45.80% (OR = 1.4580, $P < 0.1$) and 169.86% (OR = 2.6986, $P < 0.01$), respectively. This indicates that EE can improve the EI of nonpoor students more than that of poor students. Column (2) shows the results of adding individual control variables. When EE increases by one level, the possibility of starting a business increases by 49.88% (OR = 1.4988, $P < 0.1$) for poor students and 165.31% (OR = 2.6531, $P < 0.01$) for nonpoor students. Column (3) includes the results of controlling variables of individual and family factors. When EE increases by one level, the entrepreneurial possibility of poor and nonpoor students increases by 52.82% (OR = 1.5282, $P < 0.05$) and 162.00% (OR = 2.6200, $P < 0.01$), respectively. Column

**Table 10. Differences between public and private universities.**

| Variable | Dependent variable: Entrepreneurship intention | | | |
|---|---|---|---|---|
| | **(1)** | **(2)** | **(3)** | **(4)** |
| **Panel A: Public** | | | | |
| education | 1.5432*** | 1.5477*** | 1.5709*** | 1.4744** |
| | (0.2171) | (0.2196) | (0.2307) | (0.2223) |
| **Individual** | | Yes | Yes | Yes |
| **Family** | | | Yes | Yes |
| **School** | | | | Yes |
| **Constant** | 0.2609*** | 0.2020** | 0.1991 | 2.7067 |
| | (0.1248) | (0.1462) | (0.2045) | (6.5741) |
| **Observations** | 350 | 350 | 350 | 350 |
| **Pseudo R-squared** | 0.0207 | 0.0238 | 0.0705 | 0.0932 |
| **Panel B: Private** | | | | |
| education | 5.2519*** | 5.2584*** | 5.0957*** | 6.7819*** |
| | (1.7284) | (1.7746) | (1.7774) | (2.8120) |
| **Individual** | | Yes | Yes | Yes |
| **Family** | | | Yes | Yes |
| **School** | | | | Yes |
| **Constant** | 0.0027*** | 0.0074*** | 0.0011*** | 0.1042 |
| | (0.0029) | (0.0096) | (0.0022) | (0.8751) |
| **Observations** | 168 | 168 | 168 | 168 |
| **Pseudo R-squared** | 0.1730 | 0.1980 | 0.2450 | 0.3450 |
| **Group difference** | 1.2247 *** | 1.2231*** | 1.1767*** | 1.5260 *** |

Note: Regression values are odds ratios. The values in parentheses are standard errors. Differences in coefficients between groups were tested using Fisher's bootstrap based (1000 times) approach.

*** $p < 0.01$

** $p < 0.05$

* $p < 0.1$

(4) shows the results when all control variables are included. When EE increases by one level, the entrepreneurial possibility of poor students and nonpoor students increases by 51.14% (OR = 1.5114, $P < 0.1$) and 142.26% (OR = 2.4226, $P < 0.01$), respectively. The difference test value of the intergroup coefficient is significant by at least 0.1, indicating that nonpoor students have more EI than poor students after receiving EE. The empirical results support hypothesis H2d, and EE significantly affects the EI of poor and nonpoor students. According to the regression results, after receiving EE, the effect of nonpoor students on improving EI is higher than that of poor students. This shows that EE has increased the EI gap between poor and nonpoor students.

## Discussion

### Main results and research contributions

This study aimed to evaluate the overall effect of EE on the EI of college students and its influence on different college students. Based on the sample data of Chinese college students, this study empirically tested the impact of EE on EI using the logistic model. It evaluated the heterogeneity of students from different groups according to gender, household registration, university type, and poverty. In research on entrepreneurial education's effect on college students'

**Table 11. Differences between poverty and nonpoverty college students.**

| Variable | Dependent variable: Entrepreneurship intention | | | |
|---|---|---|---|---|
| | (1) | (2) | (3) | (4) |
| **Panel A: Poverty** | | | | |
| education | 1.4580* | 1.4988* | 1.5282** | 1.5114* |
| | (0.2897) | (0.3148) | (0.3245) | (0.3249) |
| Individual | | Yes | Yes | Yes |
| Family | | | Yes | Yes |
| School | | | | Yes |
| Constant | 0.2938* | 0.0987 | 0.1831 | 9.9460 |
| | (0.1936) | (0.2631) | (0.4969) | (42.7518) |
| Observations | 169 | 169 | 169 | 169 |
| Pseudo R-squared | 0.0160 | 0.0397 | 0.0471 | 0.0706 |
| **Panel B: Nonpoverty** | | | | |
| education | 2.6986*** | 2.6531*** | 2.6200*** | 2.4226*** |
| | (0.4628) | (0.4674) | (0.4643) | (0.4350) |
| Individual | | Yes | Yes | Yes |
| Family | | | Yes | Yes |
| School | | | | Yes |
| Constant | 0.0312*** | 0.1322 | 0.1160 | 4.1364 |
| | (0.0181) | (0.2407) | (0.2206) | (14.7584) |
| Observations | 349 | 349 | 349 | 349 |
| Pseudo R-squared | 0.0852 | 0.1090 | 0.1090 | 0.1280 |
| Group difference | 0.6156** | 0.5711** | 0.5391** | 0.4718* |

Note: Regression values are odds ratios. The values in parentheses are standard errors. Differences in coefficients between groups were tested using Fisher's bootstrap based (1000 times) approach.

*** $p < 0.01$

** $p < 0.05$

* $p < 0.1$

EI, previous studies lack rigorous empirical data tested on different groups of college students, and the empirical results are inconsistent. Compared with previous literature, this study mainly makes marginal contributions to the differences between rural and urban, public and private universities, and poor and nonpoor college students and provides rigorous empirical design. Specifically, while trying to control individual, family, school, and other relevant influencing factors, this study uses robustness test methods such as variable changes, sample changes, and empirical model changes to address the endogeneity problem and provide reliable empirical evidence for the study. In addition, combining human capital theory, self-efficacy theory, and planned behavior theory, the results of this study are compared with those of previous literature, and the reasons for the consistency and inconsistency are analyzed to improve and contribute to the existing research.

The conclusion of the baseline regression is consistent with the literature; EE significantly affects the EI of college students [2, 24, 35, 37, 39, 53, 54, 56, 57, 63, 68]. The results show that when relevant factors are controlled for, the possibility of students participating in entrepreneurship increases by 86.34% when EE increases by one level (see Table 3). On the one hand, this study provides a possible explanation for the inconsistent empirical results of the positive impact of EI on EI. On the other hand, the paper tries to determine the possible

reasons for the small or negative effect of EE on the EI of college students [22, 23, 46, 48, 49, 101]. The empirical analysis showed that first, with the gradual inclusion of individual, family, and school control variables, the model's pseudo R-squared increased from 0.0544 to 0.1260, and the odds ratios of EI decreased from 2.1217 to 1.8634 (see Table 3). On the one hand, it indicates the necessity of controlling for relevant influencing factors; on the other hand, it also indicates that not including individual, family, and school control variables will lead to overestimating the effect of EE on EI. This shows the necessity of controlling for related influencing factors and proves that not including individual, family, and school control variables will lead to overestimating the effect of EE on EI. Second, the robustness test shows that the sample characteristics are an essential factor affecting the results. When samples with prior EE and experience were deleted, the odds ratios of EI increased from 1.8634 to 2.0957, indicating that exposed entrepreneurship samples would reduce the effect of EE on EI (see Table 5). After deleting the sample of students whose academic performance is at the extreme value, the odds ratios of EI increase from 1.8634 to 2.0138, indicating that the students' learning ability is not controlled and the results will be biased (see Table 6).

The positive effect of EE on the EI of women is greater than that of men. For each additional level of EE, the likelihood of female and male students participating in entrepreneurship increases by 131.92% and 83.55%, respectively (see Table 8). The results are consistent with the literature suggesting that EE has a more significant effect on EI for women [51, 63, 104] but not consistent with the literature suggesting that it has a more significant effect on men [39, 53]. The reason for the inconsistent empirical results may be the reason for the sample and practical design. From the empirical process, with the gradual addition of control variables, model fitting and empirical results have significant changes. From the theoretical perspective, if EE improves the human capital level of both men and women equally through the increase of knowledge and skills, then the improvement of self-efficacy is the reason for women overcoming the congenital disadvantage [42, 52, 63, 104]. EE is more helpful for women to improve their entrepreneurial self-efficacy. Compared with men, women are more enthusiastic and serious about EE and may hope to acquire more entrepreneurial knowledge and skills through EE. The improvement of self-efficacy promotes more women's intention to start businesses. In line with scholars' views, EE serves as a tool to reduce gender inequality in entrepreneurship [52].

The positive effect of EE on the EI of urban students is greater than that of rural students. Each additional level of EE increases the likelihood of urban and rural registered students participating in entrepreneurship by 1106.59% and 55.39%, respectively (see Table 9). Urban students have more entrepreneurial support resources and environmental advantages than rural students and benefit more from EE [35, 44, 74]. Consistent with the views of the literature [44], urban college students mainly live in cities. On the one hand, cities have better educational resources and conditions than rural areas, and their level of human capital is higher. On the other hand, there are more opportunities to directly access the information, knowledge, and resources related to entrepreneurship. The proportion of entrepreneurship in urban families and circles of friends is high, and it is easier for students to obtain experience, tacit knowledge, and support for entrepreneurship. In contrast, rural students mainly live in rural areas, and it is difficult for them to access information related to entrepreneurship due to the restrictions of the living environment and social interaction. Due to the differences between urban and rural students in human capital, self-efficacy, entrepreneurial attitude, social support, and behavioral control ability, urban students have a higher effect on improving EI after receiving entrepreneurial education than rural students. Equal EE has increased the gap between urban and rural areas in entrepreneurship, and the lack of entrepreneurship policies for rural students cannot change this inequality. Notably, the empirical results of this study are inconsistent with the view that the EI of urban students is lower than that of rural students, which may

be the problem of the sample. For example, Hunady et al. (2018) believed that urban students employed in family enterprises or stable positions had lower employment intentions than rural students [35]. However, the samples of this study come from the relatively underdeveloped western regions of China. The families of urban students generally do not have family businesses, and rural students tend to look for stable jobs due to the disadvantage of their low family economic and social resources, thus showing low EI.

The positive effect of EE on the EI of students in private universities is greater than that of students in public universities. With the increase in EE level, the probability of students from private universities and public universities participating in entrepreneurship increases by 578.19% and 47.44%, respectively (see Table 10). The empirical results are consistent with those of some scholars [53, 75]. After receiving EE, the difference in talent training mode makes the students in private universities have a higher attitude and behavior control mode toward entrepreneurship than those in public universities, which is the possible reason for the difference in EI. On the one hand, compared with private universities, the lack of entrepreneurship support and the entrepreneurship atmosphere in public universities is the superficial reason [75]. The deep-seated reason is that public universities' research attributes and research result-oriented and research-oriented talent training modes hinder students' EI. On the other hand, private universities create a favorable entrepreneurial environment and provide more entrepreneurial opportunities by cultivating application-oriented talents closely related to the regional economy and vocational positions. The results of this study are inconsistent with Walter et al.'s (2013) conclusion that the research attributes of universities are positively related to students' EI based on a study of German public universities [39]. The reason may be that most of the students involved in this study set up nontechnological small enterprises, which is inconsistent with the situation of students in developed countries.

The positive effect of EE on the EI of nonpoor students is greater than that of poor students. With the addition of one level of EE, the probability of nonpoor and poor students participating in entrepreneurship increases by 142.26% and 51.14%, respectively (see Table 11). Quite a few college students are in low-income family economic situations (see Table 2), so entrepreneurship is an effective tool for poverty alleviation through education [5–17]. However, EE widens the gap and thus exacerbates the intergenerational cycle of poverty. On the one hand, nonpoor students have the advantages of family status, social capital, and an entrepreneurial environment, and they will benefit more from the same entrepreneurial education [16, 109]. On the other hand, if poor students receive the same education, they can only bear the consequences of "the strong getting stronger and the weak getting weaker," thus widening the gap with nonpoor students regarding human capital and self-efficacy and increasing the actual inequality [6].

### Theoretical implications and practical applications

This study provides empirical evidence for the practical application of the theory. By investigating the impact of EE on college students' EI, this study finds that EE has a significant positive impact on EI. Education improves students' knowledge and skills, thus increasing students' level of human capital and entrepreneurial self-efficacy [2, 20, 24, 35, 49, 54, 55, 57, 59–61, 63, 65, 66, 98]. While improving the level of human capital, EE has a more significant effect on women's EI, thus narrowing the gap between women's and men's EI. It can be found from the research results that the applied talent training mode can better play the role of EE in improving students' EI, indicating that the education mode directly related to entrepreneurship is conducive to improving students' entrepreneurial ability. However, EE increases the relative disadvantage of rural and urban students and poor and nonpoor students in human

capital, self-efficacy, social pressure, and other aspects. Based on this, empirical research provides empirical evidence for providing educational assistance policies and measures and talent training programs for disadvantaged groups.

This study has practical significance for EE and talent cultivation policy. First, it is beneficial to construct and implement policies and measures to better play the positive role of EE in promoting EI. To encourage more college students to participate in entrepreneurship, the government and educational institutions can improve the effectiveness of EE, improve the quality of EE and create an excellent entrepreneurial culture to enhance the EI of students. Second, the gap should be narrowed between the EE and EI of students of different groups and educational fairness and equality should be promoted. Attention should be given to the EE of female students, and appropriate education programs should be formulated to improve the level of human capital and self-efficacy of female students to narrow the gap between female and male EI. The government and educational institutions should increase the help of rural college students, narrow the impact of the urban–rural gap by providing financial support and psychological help, and encourage more rural college students to participate in entrepreneurship. Public universities should change the concept of attaching importance to research, neglecting practical application, and providing entrepreneurship courses and activities that combine theory and practice with improving college students' EI. According to the theory of poverty alleviation through education, entrepreneurship is conducive to college students escaping from poverty. Policy formulation and implementation should strengthen educational assistance for poor college students and provide suitable EE and entrepreneurial opportunities. For example, special subsidy funds, entrepreneurial practice opportunities, psychological assistance, and other ways can be provided to reduce the intergenerational cycle of poverty among college students through EE.

## Research limitations and future research directions

The present study has some limitations. The data are cross-sectional, and differences in different periods cannot be compared and analyzed. The research objects were limited to samples of students from two universities in China, and the sample representativeness was limited. The long-term impact of EE on employment action cannot be assessed with only data on students in college, not after graduation. There is no in-depth analysis of the differences in the EIs of different groups from social and cultural aspects.

Future research can be carried out from the following aspects. In terms of data, the sample can be expanded horizontally and vertically, and longitudinal data can be obtained by tracking the employment situation of students after graduation. In terms of research topics, it would probably be meaningful to conduct in-depth research on urban–rural factors, economic status, and school type differences and to analyze the cultural and social factors that lead to differences.

## Conclusion

Based on questionnaire data from Chinese college students, this study examines the impact of EE on college students in general and different groups. The main conclusions are as follows: First, EE has a significant positive impact on the EI of college students. After controlling for related factors, the possibility of entrepreneurship of students increased by 86.34% for each level of increase in EE. The robustness test was carried out by changing the independent variables, deleting the samples that had received EE before entering university, deleting the samples with the best and worst performance, and changing the econometric model. Second, there are significant differences in the impact of EE on the EI of different college students. After

controlling for related factors, the entrepreneurial possibility of female and male students increased by 131.92% and 83.55%, respectively, and that of rural and urban students increased by 55.39% and 1106.59%, respectively. The likelihood of starting a business increased by 47.44% and 578.19% for public and private university students and 51.14% and 142.26% for poor and nonpoor students, respectively. The study's results provide new empirical evidence for entrepreneurship and help achieve the goals of entrepreneurship, promoting economic growth and reducing poverty.

## Supporting information

**S1 Data.**
(ZIP)

## Acknowledgments

The authors would like to acknowledge the faculty and students participating in the study.

## Author Contributions

**Conceptualization:** Juan Wang.

**Data curation:** Wanli Deng.

**Formal analysis:** Wanli Deng.

**Funding acquisition:** Wanli Deng, Juan Wang.

**Investigation:** Wanli Deng.

**Methodology:** Juan Wang.

**Project administration:** Juan Wang.

**Resources:** Wanli Deng.

**Software:** Wanli Deng.

**Supervision:** Juan Wang.

**Validation:** Wanli Deng.

**Visualization:** Wanli Deng.

**Writing – original draft:** Wanli Deng.

**Writing – review & editing:** Wanli Deng.

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
