## [Decision Letter · Decision Letter 0]

24 Apr 2023

PONE-D-23-03086The effect of entrepreneurship education on the entrepreneurial intention of different college students: Gender, household registration, school type, and poverty statusPLOS ONE

Dear Dr. Deng,

Thank you for submitting your manuscript to PLOS ONE. After careful consideration, we feel that it has merit but does not fully meet PLOS ONE’s publication criteria as it currently stands. Therefore, we invite you to submit a revised version of the manuscript that addresses the points raised during the review process.

Briefly, I have a good time reading this paper, here are my comments:

1. The introduction is required to add a few more sentences to increase the strength of this article. Kindly deeply explain the research problem, objective, and novelty in the last paragraph of the Introduction section. 

2. At the end of the literature review, kindly incorporate the gap in the literature.

3. The introduction and literature review need to be revised. The introduction gives too much background, which requires only a concise paragraph. Furthermore, literature review requires not only sorting out existing relevant literature, but also reviewing the literature, and finally explaining the research contributions of this article.

4. The discussion section needs minor improvement. This can be done by comparing your results with prior studies. 

5. The conclusion also needs to be more precise. This section needs to add statistical evidence with authentic reasoning.

6. The authors need to reveal the originality of this paper better, explicitly,   by comparing this paper and some other papers published in the literature which focus on the same/similar research topic.

We look forward to receiving your revised manuscript.

Kind regards,

Roni Bhowmik, Ph.D.

Academic Editor

PLOS ONE

Journal Requirements:

Reviewers' comments:

Reviewer's Responses to Questions

**Comments to the Author**

1. Is the manuscript technically sound, and do the data support the conclusions?

Reviewer #1: Yes

Reviewer #2: Yes

Reviewer #3: Partly

2. Has the statistical analysis been performed appropriately and rigorously? 

Reviewer #1: Yes

Reviewer #2: Yes

Reviewer #3: No

3. Have the authors made all data underlying the findings in their manuscript fully available?

Reviewer #1: Yes

Reviewer #2: Yes

Reviewer #3: No

4. Is the manuscript presented in an intelligible fashion and written in standard English?

Reviewer #1: Yes

Reviewer #2: Yes

Reviewer #3: Yes

5. Review Comments to the Author

Reviewer #1: Thank you for the invitation to read and review this article, here are my views for reference.

The paper has done a lot of research and analysis work, the analysis procedure is detailed, and the conclusions also have a good practical reference role.

Reviewer #2: Dear authors,

The paper is very interesting as it deals with a timely topic. Nevertheless, it needs to be improved to become a winning paper. Then, in my opinion, the following comments can help you reach that point:

First of all, please make your abstract attractive to readers (simple sentences without any repetition) and include 2-3 sentences ready to be cited exactly as they are. In 1 paragraph, your abstract should tell the readers why the study is important (maximum 25% of the text), what you did, i.e. your methodology (maximum 25% of the text), and what you found, i.e. main research results and their major implications (50% of the text). This is very important to promote your work because of the growing trend that authors use Google search to find and cite papers based on the abstract (instead of reading the full paper).

What is the specific research stream you have found on the Plos One that can include your contribution? how does the paper push the research forward? please, be more explicit on this issue.

The research gap, theoretical contribution, necessity, and importance are not discussed well.

The sentences are not connected well; therefore, the manuscript lacks enough coherence. For instance, "The development needs of entrepreneurship make EE present a trend of rapid development [35]. EE in universities has carried out a series of activities in curriculum construction, ...". I can not follow your storytelling style well. Please read the following manuscript which can help you improve the early draft: Shepherd, D. A., & Wiklund, J. (2020). Simple rules, templates, and heuristics! An attempt to deconstruct the craft of writing an entrepreneurship paper. Entrepreneurship Theory and Practice, 44(3), 371-390.

The literature review can be improved. The following references might be useful:

- 2022. Entrepreneurship education and graduates' entrepreneurial intentions: Does gender matter? A multi-group analysis using AMOS. Technological Forecasting and Social Change, 180, 121693.

- 2021. The impact of entrepreneurial education on technology-based enterprises development: The mediating role of motivation. Administrative Sciences, 11(4), 105.

- 2022. Measuring the impact of simulation-based teaching on entrepreneurial skills of the MBA/DBA students. In Technology and Entrepreneurship Education: Adopting Creative Digital Approaches to Learning and Teaching (pp. 77-104). Cham: Springer International Publishing.

-2022. Entrepreneurial universities and social capital: The moderating role of entrepreneurial intention in the Malaysian context. The International Journal of Management Education, 20(1), 100609.

- 2022. Antecedents of entrepreneurial intentions of female undergraduate students in Bangladesh: a covariance-based structural equation modeling approach. Journal of Women's Entrepreneurship and Education, (1-2), 137-153.

- 2014. Entrepreneurial characteristics: insights from undergraduate students in Iran. International Journal of Entrepreneurship and Small Business, 21(2), 165-182.

Compare your findings with those of the others. The authors need to draw substantive conclusions from their results, suggest implications for theory and practice, and, perhaps, develop recommendations for further research in more detail.

You shall discuss the implications and limitations. The conclusion also needs to be more precise.

Best of luck!

Reviewer #3: The structure of the paper is complete and well-organized. However, the innovation and contribution are not prominent enough. The relationship between entrepreneurship education and entrepreneurial intention has been extensively studied. The perspectives of gender, poverty have been studied. Innovation is not about putting together variables that have been studied by others. Besides, research design is not rigorous. There are loopholes in the process of collecting and organizing data. Model fit has not been carried out either. Thirdly, there is not enough discussion on the proposed hypotheses. In short, the study does not yet meet the standards for public publication.

1. Literature review should not only list existing literature. The development and gaps of the literature should be clearly sorted out.

2. It lacks of necessary theoretical analysis when proposing hypotheses.

The process of proposing hypotheses is crucial. The hypotheses of this study cannot be proposed simply by summarizing the research results of others. It should be based on theoretical foundations, phenomena, and analysis. Also, theoretical foundation should be applied.

3. In the literature review, you have listed the positive or negative effects of entrepreneurship education on entrepreneurial intention. Why do you propose the positive effect in Hypothesis 1? This lacks analysis and rationality. Others also have this problem.

4.In fact, you have reviewed the heterogeneity of gender, household registration, school type, and poverty status. Why do we need to conduct research? It is related to whether the research has innovation and contributions. As you said in Introduction, the empirical results are inconsistent. Is unifying the results an innovation? Does this result have universality and typicality?

5. Why is entrepreneurial intention variable is binary? The Likert scale is usually used in the scale of intention. Why is entrepreneurship education using a five-Likert scale and entrepreneurship willingness not used? If so, you need cite relevant supporting literature.

6. Why are the Obs inconsistent? If the observation values of some samples are missing, should they be removed?

7. When conducting group regression, heterogeneity cannot be proven simply by the different coefficients of the two regression groups. Take Table 8 as an example. This result can only indicate that entrepreneurship education is significant for both men and women in terms of entrepreneurial intentions. Heterogeneity needs to be tested and Stata has commands for this.

8. What’s about the model fit in this study? What’s the value of RMSEA, NFI, CFI…etc.

6. PLOS authors have the option to publish the peer review history of their article (what does this mean?). If published, this will include your full peer review and any attached files.

Reviewer #1: No

Reviewer #2: **Yes: **Aidin Salamzadeh

Reviewer #3: No

---

## [Author Response · Author response to Decision Letter 0]

7 Jun 2023

Response to the comments of the editor and reviewers

Dear editor and reviewers,

Thank you for your letter and the reviewers' comments on our manuscript entitled "The effect of entrepreneurship education on the entrepreneurial intention of different college students: Gender, household registration, school type, and poverty status" (ID: PONE-D-23-03086). All of these comments were very helpful for revising and improving our paper. We have carefully studied these comments and made corrections, which we hope you will approve.

The changes in the revised manuscript are marked in red.

The point-to-point responses to comments from the editor and reviewers are provided below.

Response to Editor

General Comments from the Editor

Comment: Thank you for submitting your manuscript to PLOS ONE. After careful consideration, we feel that it has merit but does not fully meet PLOS ONE's publication criteria as it currently stands. Therefore, we invite you to submit a revised version of the manuscript that addresses the points raised during the review process.

Response: Thank you for your insightful comments and warm encouragement. Your comments prompted us to carry out relevant actions on the proposal, and we provide the following reasons and modifications. Changes in the manuscript are marked in red.

Specific Comments from the Editor

Comment E-1.1: The introduction is required to add a few more sentences to increase the strength of this article. Kindly deeply explain the research problem, objective, and novelty in the last paragraph of the Introduction section. 

Response E-1.1: Thank you very much for your valuable comments. We have revised the last part of the introduction by adding a few sentences to explain in depth the problem (p. 4-5, lines 83-93), objective (p. 4, lines 81-83), and novelty (p. 5-6, line 94-111;) of the research. 

Comment E-1.2: At the end of the literature review, kindly incorporate the gap in the literature. 

Response E-1.2: Thank you very much for your advice. We have incorporated the gaps that exist in the literature into the article, mainly in the following positions (p. 4, line 67-80; p. 9, line 186-196; p. 12, line 249-250; p. 13, line 268-269).

Comment E-1.3: The introduction and literature review need to be revised. The introduction gives too much background, which requires only a concise paragraph. Furthermore, literature review requires not only sorting out existing relevant literature, but also reviewing the literature, and finally explaining the research contributions of this article. 

Response E-1.3: We thank you for this constructive comment. We have revised the introduction and literature review. First, in the background section of the introduction, merge the first paragraph with the second paragraph and simplify the statement (p. 3, lines 49-60). Second, in the literature review part, the existing literature is sorted out and reviewed in detail (p. 3-4, lines 61-80), and based on the literature review, the research contribution of this paper is presented (p. 5-6, line 94-111). 

Comment E-1.4: The discussion section needs minor improvement. This can be done by comparing your results with prior studies. 

Response E-1.4: Thank you very much for your suggestion. We have improved the discussion section to compare and analyze the results with previous studies (p. 44-48, lines 771-870). 

Comment E-1.5: The conclusion also needs to be more precise. This section needs to add statistical evidence with authentic reasoning.

Response E-1.5: Thank you very much for your advice. We modify the conclusions by adding numerical values of important empirical results as statistical evidence (p. 51-52, lines 926-941).

Comment E-1.6: The authors need to reveal the originality of this paper better, explicitly, by comparing this paper and some other papers published in the literature which focus on the same/similar research topic. 

Response E-1.6: Thank you very much for your suggestion. We have compared this paper with the literature, highlighting the shortcomings of the existing literature and the originality of this paper. First, we mainly compare this paper with relevant literature in the discussion part and show the originality of this paper by comparing the similarities and differences with previous literature and analyzing the reasons (p. 43-48, lines 753-870). Second, we contrast the introduction (p. 3-4, lines 61-80) and theoretical foundations of the paper with the relevant literature in hypothesis development (p. 9, lines 186-196) to demonstrate the paper's originality. 

Response to Reviewer #1

General Comments from Reviewer #1

Comment: The paper has done a lot of research and analysis work, the analysis procedure is detailed, and the conclusions also have a good practical reference role.

Response: Thank you for your insightful comments and warm encouragement. Your comments prompted us to carry out relevant actions on the proposal, and we provide the following reasons and modifications. The changes in the revised draft are marked in red.

Specific Comments from Reviewer #1

Comment R-1.1: This paper is not innovative enough and has made little academic contribution. There are many articles on the impact of entrepreneurship education on entrepreneurial intentions, and articles on factors such as gender and economic conditions have a large number of literature. 

Response R-1.1: Thank you very much for your comments on this paper's innovativeness and research contribution. For unreadable expressions, we may not clearly show the innovativeness and research contribution of the article. We have rerevised the innovativeness and research contribution. Revised innovativeness and research contributions in the introduction (p. 3-4, lines 61-80) and discussion section (p. 43-44, lines 757-770). 

Comment R-1.2: The data sources of the paper sample are relatively concentrated and lack representativeness.

Response R-1.2: Your comments on the study sample are greatly appreciated. As you mentioned, the research samples are only from two universities, which makes the sample less representative. The main reasons for choosing this research sample and relevant explanations are as follows. First, we choose these two universities because they belong to public and private universities, respectively, and they have a high proportion of rural students and poor students, which can meet the needs of this paper's research. Second, some scholars have conducted related studies using data from one or two universities, which we illustrate in the data section by citing 10 kinds of literature that adopt data similar to that used in this study (p. 15-16, lines 326-329). Third, the lack of representation of data sources has been noted in the limitations section of the study (p. 50, line 911-915), and future approaches to this problem have been mentioned in the outlook for future research (p. 51, line 918-920). 

Comment R-1.3: Line 744 says "fill the research gap," and line 817 says "extends.... for the theory", exaggerating the significance and theoretical contribution of this study. In addition, page 17, Table 1. Definition of variables, "feconomic", "fatherduyr", " Is the word "fenterprise" misspelled? Lines 100-101, uniform format for numbers after section: Section II, Section 3. 

Response R-1.3: Thank you for pointing out the mistakes. We have corrected these errors. Firstly, statements that exaggerate the theoretical contribution of the significance of this study have been deleted or rewritten (p. 43-44, lines 757-770). In addition, the improper expression of research significance and theoretical contribution in the article has been revised (p. 48, lines 872-873). Second, we modified the definition of variables to reduce the problems of ambiguity and misunderstanding in the naming of variables. Specifically, "feconomic", "fatherduyr" and "fenterprise" are changed to "family_economic", "father_edu_year" and "father_enterprise" respectively. There are "_" between words in the variables because when using Stata 17.0 econometric analysis software for empirical analysis, Spaces between letters are not allowed (p. 16-17, Table 1; p. 21-22, Table 2; p. 25-26, Table 3). Third, we unified the format of the numbers on lines 100-101 (p. 6, lines 112-117). 

Comment R-1.4: The language of the paper needs polishing.

Response R-1.4: Thank you very much for your advice on the language. We have carefully revised the manuscript and engaged a specialized language polishing company (AJE, verification code 51D7-C42E-69FE-039B-51CP) for language revision so that it should not be a problem anymore in the language of our new re-submitting manuscript. 

Response to Reviewer #2

General Comments from Reviewer #2

Comment: The paper is very interesting as it deals with a timely topic. Nevertheless, it needs to be improved to become a winning paper. Then, in my opinion, the following comments can help you reach that point:

Response: Thank you for your insightful comments and warm encouragement. Your comments prompted us to carry out relevant actions on the proposal, and we provide the following reasons and modifications. Changes in the manuscript are marked in red.

Specific Comments from Reviewer #2

Comment R-2.1: First of all, please make your abstract attractive to readers (simple sentences without any repetition) and include 2-3 sentences ready to be cited exactly as they are. In 1 paragraph, your abstract should tell the readers why the study is important (maximum 25% of the text), what you did, i.e. your methodology (maximum 25% of the text), and what you found, i.e. main research results and their major implications (50% of the text). This is very important to promote your work because of the growing trend that authors use Google search to find and cite papers based on the abstract (instead of reading the full paper).

Response R-2.1: Thank you very much for your suggestion. We have rewritten the abstract with your valuable suggestions (p. 2, lines 24-43).

Comment R-2.2: What is the specific research stream you have found on the Plos One that can include your contribution? How does the paper push the research forward? Please, be more explicit on this issue. 

Response R-2.2: Thank you very much for your comment. I am sorry for the lack of clarity in presenting research contributions in this paper. We have identified the research contributions in the introduction (p. 5-6, lines 94-111) and discussion section (p. 43-44, lines 757-770; p. 48-50, lines 872-908) and analyzed how this paper has taken the research forward. 

Comment R-2.3: The research gap, theoretical contribution, necessity, and importance are not discussed well. 

Response R-2.3: Thank you very much for your comment. According to your suggestions, we have revised the relevant parts of this article. First, we analyze the research gap in the introduction (p. 3-4, lines 61-80). In addition, the discussion section analyzes the research gap in conjunction with the paper's empirical results (p. 43-48, lines 757-870). Second, theoretical contributions are presented in the discussion section of this paper (p. 48-49, lines 872-885). Third, the introduction discusses the study's necessity and importance (p. 3-4, lines 49-80). 

Comment R-2.4: The sentences are not connected well; therefore, the manuscript lacks enough coherence. For instance, "The development needs of entrepreneurship make EE present a trend of rapid development [35]. EE in universities has carried out a series of activities in curriculum construction...". I can not follow your storytelling style well. Please read the following manuscript, which can help you improve the early draft: 

Shepherd, D. A., & Wiklund, J. (2020). Simple rules, templates, and heuristics! An attempt to deconstruct the craft of writing an entrepreneurship paper. Entrepreneurship Theory and Practice, 44(3), 371-390.

Response R-2.4: Thank you for your suggestions and for providing this literature. First, based on the constructive literature you recommended, we have revised this paper's research framework and presentation. Second, we have carefully revised the statement you suggested to be modified "EE in universities has carried out a series of activities in curriculum construction, .….." (p. 3, lines 49-60).

Comment R-2.5: The literature review can be improved. The following references might be useful:

[1] - 2022. Entrepreneurship education and graduates' entrepreneurial intentions: Does gender matter? A multi-group analysis using AMOS. Technological Forecasting and Social Change, 180, 121693.

[2] - 2021. The impact of entrepreneurial education on technology-based enterprises development: The mediating role of motivation. Administrative Sciences, 11(4), 105.

[3] - 2022. Measuring the impact of simulation-based teaching on entrepreneurial skills of the MBA/DBA students. In Technology and Entrepreneurship Education: Adopting Creative Digital Approaches to Learning and Teaching (pp. 77-104). Cham: Springer International Publishing.

[4] -2022. Entrepreneurial universities and social capital: The moderating role of entrepreneurial intention in the Malaysian context. The International Journal of Management Education, 20(1), 100609.

[5] - 2022. Antecedents of entrepreneurial intentions of female undergraduate students in Bangladesh: a covariance-based structural equation modeling approach. Journal of Women's Entrepreneurship and Education, (1-2), 137-153.

[6]- 2014. Entrepreneurial characteristics: insights from undergraduate students in Iran. International Journal of Entrepreneurship and Small Business, 21(2), 165-182.

Response R-2.5: Thanks for providing this literature, which helped refine this paper's literature review. We carefully read the above literature and cited six of them. (a list of references in the literature of the serial number of [1] to [88], [2] to [103], [3] to [107], [4] to [100], [5] to [93], [6] to [94])

Comment R-2.6: Compare your findings with those of the others. The authors need to draw substantive conclusions from their results, suggest implications for theory and practice, and, perhaps, develop recommendations for further research in more detail. 

Response R-2.6: Thank you very much for your suggestion. First, we mainly compare the findings of this paper with others' research in the discussion section and draw substantive conclusions by comparing the similarities and differences with previous literature (p. 44-48, lines 771-870). Second, based on comparison, recommendations are made for theory and practice (p. 48-50, lines 872-908), and detailed recommendations are made for further research (p. 51, lines 918-923). 

Comment R-2.7: You shall discuss the implications and limitations. The conclusion also needs to be more precise. 

Response R-2.7: Thank you very much for your suggestion. We amend the following aspects. First, in the discussion section, we have discussed the implications of the research findings in this paper（p. 43-48, lines 771-870）. Second, we have discussed the limitations of this paper (p. 50, lines 911-917). Third, we have revised the conclusion by adding the numerical values of important empirical results as statistical evidence to make the conclusion more precise (p. 51-52, lines 926-941). 

Response to Reviewer #3

General Comments from Reviewer #3

Comment: The structure of the paper is complete and well-organized. However, the innovation and contribution are not prominent enough. The relationship between entrepreneurship education and entrepreneurial intention has been extensively studied. The perspectives of gender, poverty have been studied. Innovation is not about putting together variables that have been studied by others. Besides, research design is not rigorous. There are loopholes in the process of collecting and organizing data. Model fit has not been carried out either. Thirdly, there is not enough discussion on the proposed hypotheses. In short, the study does not yet meet the standards for public publication. 

Response: Thank you for your insightful comments and warm encouragement. In particular, your suggestions on the empirical aspects go a long way toward improving the quality of this paper. Your comments have prompted us to take relevant actions on the proposal, and we provide the following reasons and amendments. Changes in the manuscript are marked in red.

Specific Comments from Reviewer #3

Comment R-3.1: Literature review should not only list existing literature. The development and gaps of the literature should be clearly sorted out. 

Response R-3.1: Thank you very much for your comment. We revise the literature review and review the developments and gaps in the existing literature in the introduction (p.3-4, lines 61-80) and the theoretical foundations and hypothesis development section (p.8-15, lines 154-314).

Comment R-3.2: It lacks of necessary theoretical analysis when proposing hypotheses. The process of proposing hypotheses is crucial. The hypotheses of this study cannot be proposed simply by summarizing the research results of others. It should be based on theoretical foundations, phenomena, and analysis. Also, theoretical foundation should be applied. 

Response R-3.2: Thank you very much for your comment. We have modified the section in which hypotheses are proposed, and the necessary theoretical analysis is carried out when hypotheses are proposed. Based on relevant theories and previous literature, this paper analyzes the current phenomenon of entrepreneurial education on college students' entrepreneurial intention and puts forward research hypotheses (p.8-15, lines 154-314).

Comment R-3.3: In the literature review, you have listed the positive or negative effects of entrepreneurship education on entrepreneurial intention. Why do you propose the positive effect in Hypothesis 1? This lacks analysis and rationality. Others also have this problem. 

Response R-3.3: Thank you very much for your comment. You are right. Indeed, it is unreasonable to assume that entrepreneurial education positively impacts entrepreneurial intention without analysis and explanation. Moreover, in proposing the hypothesis of the heterogeneous impact of entrepreneurial education on entrepreneurial intentions of male and female students, urban and rural college students, public and private university students, and poor and non-poor students, there are positive, negative, and no impact results in the previous literature, so it is incorrect to preset the direction of the impact in advance. 

According to your valuable comments, we have revised the part that puts forward the hypothesis. First, in the hypothesis (H1) about the impact of entrepreneurial education on entrepreneurial intention, we make a detailed analysis of why the negative impact of the results of this study is difficult to appear to demonstrate why this paper proposes the hypothesis that entrepreneurial education has a positive impact on entrepreneurial intention (p. 10, line 197-210). Second, in the four hypotheses of heterogeneity analysis (H2a, H2b, H2c, and H2d), we changed the way of proposing the hypotheses and replaced the original statement of positive impact with the that there are differences in the entrepreneurial intentions of college students from different entrepreneurial education groups. In addition, analysis is given to illustrate the rationality of the proposed hypothesis (p. 10-15, lines 213-314). 

Comment R-3.4: In fact, you have reviewed the heterogeneity of gender, household registration, school type, and poverty status. Why do we need to conduct research? It is related to whether the research has innovation and contributions. As you said in Introduction, the empirical results are inconsistent. Is unifying the results an innovation? Does this result have universality and typicality? 

Response R-3.4: Your suggestion is significant. For unreadable expressions, we may not clearly show the innovativeness and research contribution of the article. Therefore, we have revised it according to your comments. First, we revised innovativeness and research contributions, mainly in the introduction (p. 3-4, lines 61-80; p. 5-6, lines 94-111) and discussion sections (p. 43-44, lines 757-770). Second, we do not presuppose empirical results (H2a, H2b, H2c, and H2d) in the hypothesis proposed by heterogeneity analysis and do not adopt the method of unifying empirical results (p. 12, line 245-246; p. 13, line 264-265; p. 14, line 291-292; p. 15, line 313-314). Moreover, in the discussion section, the empirical results of this paper are compared with the positive, negative, and no-impact results of previous literature, and the theoretical analysis is carried out (p. 44-48, lines 771-870). Thirdly, this study uses the samples of college students from universities in underdeveloped areas of China for analysis, incorporates the control variables of individuals, families, and schools, and conducts robustness tests, which makes the research results universal and typical to some extent. However, it must be admitted that whether the results of this study are consistent with those of foreign countries and developed regions in China, samples from other regions need to be added for comparative analysis, which will be the direction of future research. 

Comment R-3.5: Why is entrepreneurial intention variable is binary? The Likert scale is usually used in the scale of intention. Why is entrepreneurship education using a five-Likert scale and entrepreneurship willingness not used? If so, you need cite relevant supporting literature. 

Response R-3.5: Thank you very much for your comment. The following explains the reasons for setting the entrepreneurial variable as a binary variable and citing relevant literature. First, we set the question of entrepreneurial intention as a binary variable in the questionnaire. The main reason for using binary variables is that in the questionnaire design, binary variables are used instead of five-point Likert scales to evaluate the slight differences in whether students have entrepreneurial intentions. Secondly, in the description of the variable of entrepreneurial intention, we quote seven papers that also use binary variables to support the setting of binary variables as the variable of entrepreneurial intention (p. 17, lines 346-348). Third, thank you for your advice. Indeed, most literature sets entrepreneurial intention as a five-or seven-point Likert scale. In future questionnaires, we will add the way of designing entrepreneurial intention as a five-level Likert scale.

Comment R-3.6: Why are the Obs inconsistent? If the observation values of some samples are missing, should they be removed?

Response R-3.6: Thank you for pointing out the mistakes. We have corrected these errors. We have adapted the number of observations to be consistent by deleting samples with missing values. After deleting the samples with missing values, the samples change from 564 to 518 (p. 16, lines 336-337). Tables 2-11 show the sample results with missing values deleted, and the number of observations in each group of regressions is the same in each table.

Comment R-3.7: When conducting group regression, heterogeneity cannot be proven simply by the different coefficients of the two regression groups. Take Table 8 as an example. This result can only indicate that entrepreneurship education is significant for both men and women in terms of entrepreneurial intentions. Heterogeneity needs to be tested and Stata has commands for this. 

Response R-3.7: Thank you for pointing out the mistakes. We have corrected these errors and reprogrammed the interpretation of the regression results of the heterogeneity analysis. First, the coefficient difference between groups is tested. Differences in coefficients between groups were tested using Fisher Bootstrap and Permutation test, and differences between groups were tested using the "bdiff" command of Stata software with 1000 replications of put-back sampling. Tables 8-11 show the coefficients and significance of the differences between groups. Second, the interpretation of the results of the heterogeneity analysis was rewritten. Indeed, the logistic model is used in the type variable regression analysis, and heterogeneity cannot be proved simply by the different coefficients of the two regression groups. Therefore, we deleted all the explanations of the results of the heterogeneity analysis in Tables 8-11 and rewrote the explanations of the empirical results (p. 35-36, line 633-659; p. 37-38, line 666-690; p. 39-40, line 695-722; p. 41-42, line 728-748). 

Comment R-3.8: What's about the model fit in this study? What's the value of RMSEA, NFI, CFI…etc. 

Response R-3.8: Thank you for your comments. We have added relevant tests regarding model fit and show the fitted values (Pseudo R-squared) in Tables 3-11. In addition, indicators such as RMSEA, NFI, and CF are not listed in the table because they may not apply to the logit model. 

---

## [Decision Letter · Decision Letter 1]

19 Jun 2023

PONE-D-23-03086R1The effect of entrepreneurship education on the entrepreneurial intention of different college students: Gender, household registration, school type, and poverty statusPLOS ONE

Dear Dr. Deng,

Thank you for submitting your manuscript to PLOS ONE. After careful consideration, we feel that it has merit but does not fully meet PLOS ONE’s publication criteria as it currently stands. Therefore, we invite you to submit a revised version of the manuscript that addresses the points raised during the review process.I have read the comments of reviewers, your responses to the comments, and your updated manuscript thoroughly.You still need to respond to the minor comments/suggested revisions to improve your work.Please submit your revised manuscript by Aug 03 2023 11:59PM. If you will need more time than this to complete your revisions, please reply to this message or contact the journal office at plosone@plos.org. Please include the following items when submitting your revised manuscript:A rebuttal letter that responds to each point raised by the academic editor and reviewer(s). You should upload this letter as a separate file labeled 'Response to Reviewers'.A marked-up copy of your manuscript that highlights changes made to the original version. You should upload this as a separate file labeled 'Revised Manuscript with Track Changes'.An unmarked version of your revised paper without tracked changes. You should upload this as a separate file labeled 'Manuscript'.If applicable, we recommend that you deposit your laboratory protocols in protocols.io to enhance the reproducibility of your results. Protocols.io assigns your protocol its own identifier (DOI) so that it can be cited independently in the future. For instructions see: https://journals.plos.org/plosone/s/submission-guidelines#loc-laboratory-protocols. Additionally, PLOS ONE offers an option for publishing peer-reviewed Lab Protocol articles, which describe protocols hosted on protocols.io. Read more information on sharing protocols at https://plos.org/protocols?utm_medium=editorial-email&utm_source=authorletters&utm_campaign=protocols.

We look forward to receiving your revised manuscript.

Kind regards,

Roni Bhowmik, Ph.D.

Academic Editor

PLOS ONE

Journal Requirements:

Reviewers' comments:

Reviewer's Responses to Questions

**Comments to the Author**

1. If the authors have adequately addressed your comments raised in a previous round of review and you feel that this manuscript is now acceptable for publication, you may indicate that here to bypass the “Comments to the Author” section, enter your conflict of interest statement in the “Confidential to Editor” section, and submit your "Accept" recommendation.

Reviewer #2: All comments have been addressed

Reviewer #3: All comments have been addressed

Reviewer #4: All comments have been addressed

2. Is the manuscript technically sound, and do the data support the conclusions?

Reviewer #2: Yes

Reviewer #3: Yes

Reviewer #4: Yes

3. Has the statistical analysis been performed appropriately and rigorously? 

Reviewer #2: Yes

Reviewer #3: Yes

Reviewer #4: Yes

4. Have the authors made all data underlying the findings in their manuscript fully available?

Reviewer #2: Yes

Reviewer #3: Yes

Reviewer #4: Yes

5. Is the manuscript presented in an intelligible fashion and written in standard English?

Reviewer #2: Yes

Reviewer #3: Yes

Reviewer #4: Yes

6. Review Comments to the Author

Reviewer #2: You have put a reasonable effort into minimizing the mistakes and improving the paper substantially. The new version of the paper has added two other parts to enhance the theoretical parts while trying to expand the previous version’s theoretical underpinnings and practical implications. In the current version, I can see justifications for how the selected theoretical stance can be an appropriate lens for studying the issue in question. Given the level of effort that has been put into the new version of the paper and due to the paper’s relevance to the topic of the issue, I am satisfied with the current version of the article.

Reviewer #3: (No Response)

Reviewer #4: The paper is clearly structured. From the overall presentation I would say that interesting research work has been done. The topic is also important for the readers of the journal. In the revised version, the manuscript has been extended and improved

The dependent variable is the students' EI (entrepreneurial intention). Therefore, entrepreneurial intention has to be included as dependent variable in all Tables (not “Entrepreneurship”).

7. PLOS authors have the option to publish the peer review history of their article (what does this mean?). If published, this will include your full peer review and any attached files.

Reviewer #2: **Yes: **Aidin Salamzadeh

Reviewer #3: **Yes: **Xiaohong Ma

Reviewer #4: No

---

## [Author Response · Author response to Decision Letter 1]

20 Jun 2023

Response to the comments of the editor and reviewers

Dear editor and reviewers,

Thank you for your letter and the reviewers' comments on our manuscript entitled "The effect of entrepreneurship education on the entrepreneurial intention of different college students: Gender, household registration, school type, and poverty status" (ID: PONE-D-23-03086R1). All of these comments were very helpful for revising and improving our paper. We have carefully studied these comments and made corrections, which we hope you will approve.

The changes in the revised manuscript are marked in red.

The point-to-point responses to comments from the editor and reviewers are provided below.

Response to Journal Requirements

Comment: Please review your reference list to ensure that it is complete and correct. If you have cited papers that have been retracted, please include the rationale for doing so in the manuscript text, or remove these references and replace them with relevant current references. Any changes to the reference list should be mentioned in the rebuttal letter that accompanies your revised manuscript. If you need to cite a retracted article, indicate the article's retracted status in the References list and also include a citation and full reference for the retraction notice.

Response: We have carefully checked to ensure that the list of references in this paper is complete and correct. First, this paper does not cite the retracted papers. Second, compared with the first submission, seven kinds of literature were added to the first revised draft (ID: PONE-D-23-03086R1), while no literature was added to this revised draft. Third, the list of seven additional papers is as follows (note: the numbers in [] are the serial numbers of the papers in the article) :

[88] Dana L-P, Tajpour M, Salamzadeh A, Hosseini E, Zolfaghari M. The impact of entrepreneurial education on technology-based enterprises development: The mediating role of motivation. Administrative Sciences. 2021;11(4):105.

[93] Salamzadeh A, Tajpour M, Hosseini E. Measuring the impact of simulation-based teaching on entrepreneurial skills of the MBA/DBA students. Technology and Entrepreneurship Education: Adopting Creative Digital Approaches to Learning and Teaching: Springer; 2022. p. 77-104.

[94] Salamzadeh Y, Sangosanya TA, Salamzadeh A, Braga V. Entrepreneurial universities and social capital: The moderating role of entrepreneurial intention in the Malaysian context. International Journal of Management Education. 2022;20(1). doi: 10.1016/j.ijme.2022.100609. PubMed PMID: WOS:000760340200006.

[100] Salamzadeh A, Farjadian AA, Amirabadi M, Modarresi M. Entrepreneurial characteristics: Insights from undergraduate students in Iran. International Journal of Entrepreneurship Small Business. 2014;21(2):165-82.

[103] Rahman MM, Salamzadeh A, Tabash M. Antecedents of entrepreneurial intentions of female undergraduate students in Bangladesh: A covariance-based structural equation modeling approach. Journal of Women's Entrepreneurship and Education. 2022;(1-2):137-53. doi: 10.28934/jwee22.12.pp137-153.

[107] Ramadani V, Rahman MM, Salamzadeh A, Rahaman MS, Abazi-Alili H. Entrepreneurship education and graduates' entrepreneurial intentions: Does gender matter? A multi-group analysis using AMOS. Technological Forecasting Social Change. 2022;180:121693. 

[110] Assari S. Parental Education Attainment and Educational Upward Mobility; Role of Race and Gender[J]. Behavioral Sciences, 2018, 8(11).

Response to Reviewer #2

Comment 2: You have put a reasonable effort into minimizing the mistakes and improving the paper substantially. The new version of the paper has added two other parts to enhance the theoretical parts while trying to expand the previous version's theoretical underpinnings and practical implications. In the current version, I can see justifications for how the selected theoretical stance can be an appropriate lens for studying the issue in question. Given the level of effort that has been put into the new version of the paper and due to the paper's relevance to the topic of the issue, I am satisfied with the current version of the article.

Response 2: Thank you for your insightful comments and warm encouragement. 

Response to Reviewer #4

Comment 4.1: The paper is clearly structured. From the overall presentation I would say that interesting research work has been done. The topic is also important for the readers of the journal. In the revised version, the manuscript has been extended and improved. 

Response 4.1: Thank you for your insightful comments and warm encouragement. 

Comment 4.2: The dependent variable is the students' EI (entrepreneurial intention). Therefore, entrepreneurial intention has to be included as dependent variable in all Tables (not "Entrepreneurship"). 

Response 4.2: Thank you for your careful advice. We have changed the dependent variable in all tables to entrepreneurial intention. Changes in the manuscript are marked in red.

---

## [Decision Letter · Decision Letter 2]

5 Jul 2023

The effect of entrepreneurship education on the entrepreneurial intention of different college students: Gender, household registration, school type, and poverty status

PONE-D-23-03086R2

Dear Dr. Deng,

We’re pleased to inform you that your manuscript has been judged scientifically suitable for publication and will be formally accepted for publication once it meets all outstanding technical requirements.

Kind regards,

Roni Bhowmik, Ph.D.

Academic Editor

PLOS ONE

Additional Editor Comments (optional):

Reviewers' comments:

Reviewer's Responses to Questions

**Comments to the Author**

1. If the authors have adequately addressed your comments raised in a previous round of review and you feel that this manuscript is now acceptable for publication, you may indicate that here to bypass the “Comments to the Author” section, enter your conflict of interest statement in the “Confidential to Editor” section, and submit your "Accept" recommendation.

Reviewer #3: All comments have been addressed

Reviewer #4: All comments have been addressed

2. Is the manuscript technically sound, and do the data support the conclusions?

Reviewer #3: Yes

Reviewer #4: (No Response)

3. Has the statistical analysis been performed appropriately and rigorously? 

Reviewer #3: Yes

Reviewer #4: (No Response)

4. Have the authors made all data underlying the findings in their manuscript fully available?

Reviewer #3: Yes

Reviewer #4: (No Response)

5. Is the manuscript presented in an intelligible fashion and written in standard English?

Reviewer #3: Yes

Reviewer #4: (No Response)

6. Review Comments to the Author

Reviewer #3: (No Response)

Reviewer #4: (No Response)

7. PLOS authors have the option to publish the peer review history of their article (what does this mean?). If published, this will include your full peer review and any attached files.

Reviewer #3: **Yes: **Xiaohong Ma

Reviewer #4: No

---

## [Editor Report · Acceptance letter]

11 Jul 2023

PONE-D-23-03086R2 

The effect of entrepreneurship education on the entrepreneurial intention of different college students: Gender, household registration, school type, and poverty status 

Dear Dr. Deng:

I'm pleased to inform you that your manuscript has been deemed suitable for publication in PLOS ONE. Congratulations! Your manuscript is now with our production department. 

Kind regards, 

on behalf of

Associate Professor Roni Bhowmik 

Academic Editor

PLOS ONE